# HMARL-CBF – Hierarchical Multi-Agent Reinforcement Learning with Control Barrier Functions for Safety-Critical Autonomous Systems

**H. M. Sabbir Ahmad**[1], **Ehsan Sabouni**[1], **Alexander Wasilkoff**[1], **Param Budhraja**[1],
**Zijian Guo**[1] **Songyuan Zhang**[2], **Chuchu Fan**[2], **Christos Cassandras**[1], **Wenchao Li**[1]
[1]Boston University, [2]Massachusetts Institute of Technology

{sabbir92,esabouni,awasilkoff,paramb,zjguo,cgc,wenchao}@bu.edu
{szhang21,chuchu}@mit.edu

## Abstract

We address the problem of safe policy learning in multi-agent safety-critical autonomous systems. In such systems, it is necessary for each agent to meet the safety requirements at all times while also cooperating with other agents to accomplish the task. Toward this end, we propose a safe Hierarchical Multi-Agent Reinforcement Learning (HMARL) approach based on Control Barrier Functions (CBFs). Our proposed hierarchical approach decomposes the overall reinforcement learning problem into two levels — learning joint cooperative behavior at the higher level and learning safe individual behavior at the lower or agent level, conditioned on the high-level policy. Specifically, we propose a skill-based HMARL-CBF algorithm in which the higher-level problem involves learning a joint policy over the skills for all the agents, and the lower-level problem involves learning policies to execute the skills safely with CBFs. We validate our approach in challenging environment scenarios, whereby a large number of agents have to safely navigate through conflicting road networks. Compared with existing state-of-the-art methods, our approach significantly improves the safety, achieving a near-perfect ($\geq 95\%$) success/safety rate while improving performance across all the environments [1].

## 1 Introduction

Safety-critical multi-agent systems are rapidly expanding across diverse domains, including self-driving cars [37, 90], unmanned aerial vehicles (UAVs) [84, 54], swarm robotics [41], soft robotics [53, 72, 31], and autonomous underwater vehicles (AUVs) [55]. As these systems operate in complex and often unpredictable environments, failures can lead to catastrophic consequences—endangering human lives, causing environmental damage, or resulting in severe economic losses. Hence, it is crucial to design cooperative control policies that enable agents to achieve their objectives while ensuring safety with respect to both other agents and the environment [83, 70].

Learning a flat policy using existing Multi-Agent Reinforcement Learning (MARL) [40, 57, 64] algorithms by adjoining the safety constraints to the reward function provides one possible approach. However, these approaches suffer from scalability and high sample complexity as the number of agents grows, and most importantly, they lack formal safety guarantees. The problem is further exacerbated in partially observable settings. Safety in both single- and multi-agent reinforcement learning has been studied within the constrained Markov Decision Process (CMDP) framework [89, 2, 22]. In this setting, constraints are typically expressed as expectations of cumulative (discounted) costs over entire trajectories, thereby ensuring safety only in an average or statistical sense on trajectories.

---

[1]Project website: https://bu-depend-lab.github.io/HMARL-CBF/

However, safety-critical systems necessitate *pointwise-in-time* guarantees—i.e., constraints that must hold at every time step along each trajectory—rather than merely in expectation over trajectories.

A hierarchical approach, as proposed in [67, 71, 20], can alleviate sample complexity, but existing methods do not tackle safety. To bridge these gaps, we propose a hierarchical multi-agent reinforcement learning framework based on CBFs. The higher-level policy coordinates cooperative behaviors across agents, while the lower-level policy enforces safety through CBFs. The high-level and low-level policies are trained jointly to enable safe cooperation at scale. We summarize our main contributions below:

1. We introduce **HMARL-CBF**, a novel centralized-training–decentralized-execution (CTDE) hierarchical multi-agent reinforcement learning framework that guarantees safety using Control Barrier Functions (CBFs). This approach enforces pointwise-in-time safety constraints throughout both training and execution across each trajectory.

2. Our hierarchical structure leverages skill-based decomposition, where the high-level policy coordinates cooperative behaviors across agents, while the low-level policy learns and executes safe skills guided by CBF-based safety constraints. This design guarantees safety both during training and in real-world deployment.

3. We validate our proposed approach on challenging conflicting environment scenarios where a large number of agents must each travel safely from their origin to their destination without colliding with other agents. Simulation results demonstrate superior performance and safety compliance, achieving a near-perfect ($\geq 95\%$) success/safety rate compared to existing benchmark methods.

## 2   Related Works

**Hierarchical Multi-Agent RL (HMARL).** Hierarchical RL structures decision-making hierarchically, accelerating learning [51]. Classical methods include HAM [48], MAXQ [15], options [65, 52], and feudal architectures [13]. Deep HRL extends this with subgoal-based [69, 44], option-based [5, 29], and skill-based frameworks [36, 19]. HMARL introduces hierarchy in MARL for enhanced coordination and exploration. Notable methods include value decomposition (QTRAN [63], MAXQ [15], HAVEN [71], [20]), feudal structures [1], and option-based hierarchies [68]. Temporal abstraction is explored in [67], while meta-policy learning appears in [69]. Skill discovery in HRL has explored mutual information maximization [21], trajectory clustering [43], and hierarchical skill identification [79], option learning [28], imitation [61], and adaptive refinement via Skill-Critic [27]. However, these works define skills as fixed-length or variable-length sequences that lack interpretable semantics and safety guarantees, limiting applicability to safety-critical systems.

**Safe RL.** Safety in multi-agent systems is commonly addressed using the Constrained Markov Decision Process (CMDP) framework [89, 2, 22, 88]. Approaches include primal methods [10, 11, 39] and primal-dual methods [38, 16, 32]. While CMDPs ensure safety over trajectories, they do not consider pointwise time constraints critical for safety-critical systems. Multi-agent scenarios further complicate this, as the number of constraints scales with agents. Primal-dual methods [11, 73, 23] also face issues with training instability, slow convergence, and hyperparameter sensitivity. Additionally, works addressing reach-avoid problems [62, 82, 87] are primarily limited to single-agent fully observable settings. Previous works on RL-CBFs have harnessed exact or approximate models [59, 6, 8] to learn safe policies. The existing methods tackle safety for single-agent systems [8, 17, 50], and multi-agent systems [18]. [59] considers a multi-agent cooperative setting conditioned on a fixed coordination policy, which can be suboptimal. In [8], solved constrained RL problem with CBF-based chance constraints using the augmented Lagrangian, which nullifies the merits of CBF filters. Additionally, existing works require a nominal policy which is learned [17, 7], however, this can be challenging for MAS. Finally, [86, 85] proposed methods for multi-agent systems, but such methods suffer from high sample complexity and scalability due to the flat policy structure.

## 3   Background

### 3.1   Semi-Markov Decision Process

A Semi-Markov Decision Process (SMDP) is a generalization of a MDP that allows actions to take variable amounts of time steps before transitioning to the next state. Building on that, similar to [42],

we model a Multi-Agent Semi-Markov Decision Process (MSMDP) as a tuple $(\mathcal{N}, \mathcal{S}, \mathcal{Z}, P, R, \gamma, \mathcal{T})$ described as follows: $\mathcal{N}$ is a finite set of $N$ agents, with each agent $i \in \mathcal{N}$ having an individual action set $\mathcal{Z}^i$. The joint action space $\mathcal{Z} = \prod_{i=1}^{n} \mathcal{Z}^i$ consists of joint actions $z = \langle z^1, \ldots, z^n \rangle$, representing the joint actions of the agents. The joint state space is denoted by $\mathcal{S} = \prod_{i \in \mathcal{N}} S^i \times \mathcal{S}^e$ where $\mathcal{S}^e$ is the environment state where $\mathcal{S}^i$ is the state space of agent $i$. The reward function is denoted by $R : \mathcal{S} \times \mathcal{Z} \times \mathbb{N} \to \mathbb{R}$ where $R(s, z, k)$ corresponds to the reward accrued executing action $z$ in state $s$ in $k$ steps. $\gamma$ is the discount factor, and the multistep transition probability function $P : \mathcal{S} \times \mathbb{N} \times \mathcal{S} \times \mathcal{Z} \to [0, 1]$ gives the probability of transitioning from state $s$ to state $s'$ in $k$ time steps as a result of the joint action $z$. Since the joint actions may have varying termination times, $P$ is influenced by the duration of executing an action from the state it is initiated until transitioning to the next state (also referred to as the decision epoch) and a termination scheme $\mathcal{T}$.

Three termination strategies for the temporally extended joint actions, $\tau_{\text{any}}$, $\tau_{\text{all}}$, and $\tau_{\text{continue}}$, are analyzed in [58, Figure 1]. First, we define the decision epoch as the time when either a subset or all the agents select new actions. Synchronous schemes, $\tau_{\text{any}}$ and $\tau_{\text{all}}$, set decision epochs when either any or all actions within a joint action terminate, with all agents selecting new actions. In contrast, the asynchronous scheme $\tau_{\text{continue}}$ sets the decision epoch when any of the actions within the joint action terminate. Then let those actions that did not terminate naturally continue running while initiating new actions for the agents that completed their action. Our decentralized approach adopts this asynchronous action update strategy.

The equation corresponding to the state value function $V^\pi(s)$ and state-action value function $Q^\pi(s, z)$ are defined below for a MSMDP:

$$V^\pi(s) = \mathbb{E}_{\boldsymbol{\pi}}\left[Q^\pi(s, z)\right] = \mathbb{E}_{z, k, s'}\left[R(s, z, k) + \gamma^k V^\pi(s')\right] \tag{1}$$

$$Q^\pi(s, z) = \mathbb{E}_{k, s'}\left[R(s, z, k) + \gamma^k V^\pi(s')\right] = \mathbb{E}_{k, s'}\left[R(s, z, k) + \gamma^k \mathbb{E}_{z'}\left[Q^\pi(s', z')\right]\right] \tag{2}$$

where $k$ represents the time interval until agent(s) update their action, and $R(s, z, k)$ represents the reward obtained by taking the joint action $z$ in state $s$ over the random time interval $k$. The learning objective is to find the joint policy $\pi$ for all the agents that maximizes: $\mathbb{E}_\pi\left[\sum_{t=0}^{\infty} \gamma^{\sum_{t'=0}^{t-1} k_{t'}} R(s_t, z_t, k_t)\right]$.

## 3.2 Control Barrier Functions

Consider the following agent dynamics (with the agent index dropped for simplicity):

$$\dot{s} = f(s) + g(s)a, \tag{3}$$

where $s \in \mathbb{R}^n$ is the state of the agent, $a \in \mathcal{A} \subseteq \mathbb{R}^q$ are the primitive actions, and $f : \mathbb{R}^n \to \mathbb{R}^n$ and $g : \mathbb{R}^n \to \mathbb{R}^{n \times q}$ are locally Lipschitz.

**Definition 3.1** (Class $\mathcal{K}$ function). A continuous function $\alpha : [0, \beta) \to [0, \infty], \beta > 0$ is said to belong to class $\mathcal{K}$ if it is strictly increasing and $\alpha(0) = 0$.

**Definition 3.2.** A set $C$ is forward invariant for system (3) if for every $s(0) \in C$, we have $s(t) \in C$, for all $t \geq 0$.

**Definition 3.3** (Control barrier function [3]). Given a continuously differentiable function $b : \mathbb{R}^n \to \mathbb{R}$ and the set $C := \{s \in \mathbb{R}^n : b(s) \geq 0\}$, $b(s)$ is a candidate control barrier function (CBF) for the system (3) if there exists a class $\mathcal{K}$ function $\alpha$ such that

$$\sup_{a \in \mathcal{A}}\left[L_f b(s) + L_g b(s)a + \alpha(b(s))\right] \geq 0, \tag{4}$$

for all $s \in C$, where $L_f, L_g$ denote the Lie derivatives along $f$ and $g$, respectively.

**Theorem 3.4** ([3]). *Given a constraint $b(s(t))$ with the associated set $C$, any Lipschitz continuous controller $a(t)$, that satisfies (4) $\forall t \geq t_0$ renders the set $C$ forward invariant for the control system in (3).*

**Definition 3.5** (Control Lyapunov function (CLF) [3]). A continuously differentiable function $V : \mathbb{R}^n \to \mathbb{R}$ is a globally and exponentially stabilizing CLF for (3) if there exists constants $c_i \in \mathbb{R}_{>0}$, $i = 1, 2$, such that $c_1 \|s\|^2 \leq V(s) \leq c_2 \|s\|^2$, and the following inequality holds

$$\inf_{a \in \mathcal{A}}\left[L_f V(s) + L_g V(s)a + \eta(s)\right] \leq e, \tag{5}$$

where $e$ makes this a soft constraint.

Additional details on Higher Order CBFs (HOCBFs) as well relaxation of the affine dynamics assumption are included in Appendix A.2.

## 4    Problem Formulation

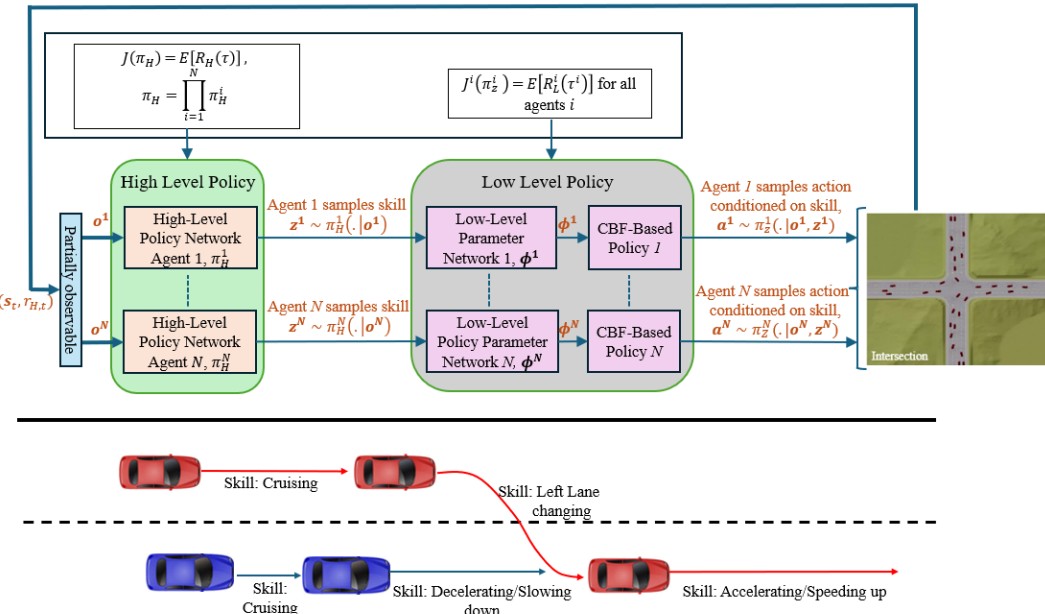

Figure 1: At the higher level, agents choose skills in a decentralized manner based on $\pi_H$ conditioned on their own observation, whereas at the lower level every agent $i$ executes these skills by learning a parametric CBF-based policy $pi_{z^i}$ conditioned on their observation and the skill. The extrinsic trajectory return $R_H(\tau)$ is used to jointly learn the cooperative policy for all the agents centrally, and the intrinsic trajectory return $R_L^i(\tau^i)$ for agent $i$'s trajectory $\tau^i$ is used to learn the policies corresponding to the skills. The high-level and low-level policy networks are shared among all agents.

We consider a cooperative multi-agent setting among a set of agents. Let the set of agents be denoted by $\mathcal{N} = \{1, \ldots, N\}$. We consider that every agent $i \in \mathcal{N}$ has dynamics as in (3). We partition the state space into two disjoint spaces corresponding to the agent states and the environment states. The environment state is denoted by $s^e \in \mathcal{S}^e$. The state of an agent $i$ is denoted by $s^i \in \mathcal{S}^i \subset \mathbb{R}^n$ and the control input vector is denoted by $a^i \in \mathcal{A}^i = [a_{min}, a_{max}] \subset \mathbb{R}^q$ with $\mathcal{A}^i$ the control input constraint set for the agent $i$, and $a_{min}, a_{\max} \in \mathbb{R}^q$. Then, the joint state space is denoted by $\mathcal{S} = \prod_{i \in \mathcal{N}} \mathcal{S}^i \times \mathcal{S}^e$, and the joint action space of the agents is denoted by $\mathcal{A} = \prod_{i \in \mathcal{N}} \mathcal{A}^i$. In our setting, there is a performance-related cost function $l(s, a)$ associated with the system specification, where $s \in \mathcal{S}$ (more specifically $\prod_{i \in \mathcal{N}} \mathcal{S}^i$) and $a \in \mathcal{A}$. Without loss of generality, in our cooperative setting, the stage cost is defined in the form $l(s, a) = \sum_{i \in \mathcal{N}} l^i(s^i, a^i)$, where $l^i(s^i, a^i)$ denotes the stage cost associated with agent $i \in \mathcal{N}$.

We consider a partially observable setting where each agent perceives only a subset of the global state. Let $\mathcal{O}^i \subseteq \mathcal{S}$ denote the observation space of agent $i$, and the joint observation space is $\mathcal{O} = \prod_{i \in \mathcal{N}} \mathcal{O}^i$. Associated with every agent $i$ we define a set of safety-related functions $\{b_j^i\}_{j \in \mathcal{N}^i(s)}$ where $b_i^j : \mathcal{O}^i \to \mathbb{R}$, and $\mathcal{N}^i(s)$ is the index set of the constraints dependent on the state. Our safety-constrained problem is expressed as a Stochastic Optimal Control Problem (SOCP) as follows:

$$\min_{\pi \in \Pi} \mathbb{E}_{\tau \sim \pi} \left[ \sum_{t=0}^{\infty} \gamma^t l(s_t, a_t) \right]$$

$$\text{subject to} \quad \mathbb{E}_{s_t \sim p^\pi} \left[ \min\{0, b_j^i(o_t^i)\} | o_t^i = g(s_t) \right] \geq 0, \forall i \in \mathcal{N}, \; j \in \mathcal{N}^i(s_t), \; \forall t \geq 0 \quad (6)$$

where $\pi(.|s) \in \Pi$ is the joint state feedback policy of the agents, i.e., $\pi : \mathcal{S} \to \Delta(\mathcal{A})$, where $\Delta$ is the space of distributions over the support of the random variable, and $g$ maps state to agent's observation. It is noteworthy that this problem requires that the constraints are satisfied pointwise in time for every trajectory generated by $\pi$. The observation comprises of the agent's own state, states of the observable agents and the environment states. Hence, the constraints can be alternatively expressed

in pairs as $b_j^i(\boldsymbol{s}^i, \boldsymbol{s}^j) \geq 0$ where $\boldsymbol{s}^j$ is the state of another agent, or the environment. We assume that the constraints are initially satisfied upon their introduction.

**Cooperative setting with pointwise-in-time hard constraints vs. CMDP setting:** As shown in [89], CMDPs address safety in fully cooperative multi-agent settings where agents share a common reward and seek to minimize the expected cumulative cost:

$$\min_{\pi} \mathbb{E}_{\tau \sim \pi} \left[ \sum_{t=0}^{\infty} \gamma^t l\left(\boldsymbol{s}_t, \boldsymbol{a}_t\right) \right] \quad \text{subject to} \quad \mathbb{E}_{\tau \sim \pi} \left[ \sum_{t=0}^{\infty} \gamma^t b_j^i\left(\boldsymbol{s}_t, \boldsymbol{a}_t^i\right) \right] \leq \alpha_j^i, ; \forall j \tag{7}$$

While structurally similar to pointwise-in-time constraints in the single-agent setting [73], the multi-agent CMDP formulation requires policies to depend on the joint state $\boldsymbol{s}_t$, making decentralized execution infeasible in partially observable environments and linearly increasing constraint complexity with the number of agents. Our formulation overcomes these limitations by enabling decentralized safety guarantees in partially observable, safety-critical multi-agent systems.

# 5  Safe Hierarchical Reinforcement Learning (HMARL-CBF)

We propose a novel Safe Hierarchical Multi-Agent Reinforcement Learning (HMARL) approach illustrated in Figure 1 to tackle the aforementioned problem in (6). Specifically, we decompose the problem in (6) into a bi-level RL problem. The higher-level problem objective concerns the optimization of the joint performance of the agents. The lower-level optimization problem addresses the safety of the agents defined using the second term in the stage cost in (10). Our proposed HMARL method is based on the idea of skills, which is defined as follows.

**Safe agent skills.** Inspired by [66, 12], we define a safety-constrained skill $z^i$ for an agent $i$ as a seven-tuple: $(\mathcal{I}_z^i, \mathcal{J}_z^i(\boldsymbol{o}_{z,init}^i, s_{init}^e), T_{max}, \Phi_z^i, \mathcal{N}^i(\boldsymbol{o}^i), r_z^i(\boldsymbol{s}^i, \boldsymbol{a}^i), \pi_z^i)$ where $\mathcal{I}_z^i \subset \mathcal{S}^i$ is the initiation set for the skill respectively, $\boldsymbol{o}_{z,init}^i \in \mathcal{I}_z^i$ and $\boldsymbol{s}_{init}^e \in \mathcal{S}^e$ are the observation of agent $i$ and environment state at the time of initiation of the skill, $\mathcal{J}_z^i(\boldsymbol{o}_{init}^i, \boldsymbol{s}_{init}^e) \subseteq \mathcal{O}^i$ is the termination set of the skill, and $\Phi_z^i : \mathcal{O}^i \times \mathbb{N} \to \{0, 1\}$ is the termination function as defined below:

$$\Phi_z^i(\boldsymbol{o}^i, t) = \begin{cases} 1, & \text{if } \boldsymbol{o}^i \in \mathcal{J}_z^i(\boldsymbol{o}_{z,init}^i, \boldsymbol{s}_{init}^e) \text{ or}, t \geq T_{\max} \\ 0, & \text{otherwise} \end{cases} \tag{8}$$

A skill is terminated as a function of the initiation state of the skill $s_z^i$ in one of the two ways: i. either upon observing that the skill has been executed or, ii. interruption due to a timeout for exceeding the maximum allowed duration of execution, $T_{\max} \in \mathbb{N}$. $\mathcal{N}^i(\boldsymbol{o}^i)$ is the set of safety constraints for agent $i$ that has to be satisfied during the execution of the skill. We define a reward function $r_z^i(\boldsymbol{s}^i, \boldsymbol{a}^i)$ corresponding to action $\boldsymbol{a}^i$ in state $\boldsymbol{s}^i$ which is used to learn the skill. Finally, $\pi_z^i : \mathcal{O}^i \times \mathcal{J}_z^i \to \Delta(\mathcal{A}^i)$ is the safe skill policy expressed as a SOCP as follows:

$$\max_{\pi_z^i, k} \mathbb{E}_{\boldsymbol{s}_t^i, \boldsymbol{a}_t^i \sim \pi_z^i, k} \left[ \sum_{t=0}^{k} r_z^i(\boldsymbol{s}_t^i, \boldsymbol{a}_t^i) | \boldsymbol{s}_0^i = \boldsymbol{s}_{init}^i \right] \tag{9}$$

$$\text{subject to } \mathbb{E}_{\boldsymbol{s}_t}[\min\{0, b_j^i(g(\boldsymbol{s}_t))\}] \geq 0, \ \boldsymbol{s}_k^i \in \mathcal{J}_z^i, \ \forall j \in \mathcal{N}(\boldsymbol{s}_t), \quad \forall t = 0, \dots, k;$$

The set of skills for an agent $i$ is defined as $\mathcal{Z}^i$, and the joint set of skills of the $N$ agents is defined as $\mathcal{Z} = \cup_{i \in \mathcal{N}} \mathcal{Z}^i$. We define a high-level policy for the $N$ agents over their set of skills denoted by $\pi_H : \mathcal{O} \to \Delta(\mathcal{Z})$ where $\pi_H \in \Pi_H$, which becomes the high-level RL problem. Subsequently, we define $\pi_{\boldsymbol{z}} : \mathcal{O} \times \mathcal{J}_{\boldsymbol{z}} \to \Delta(\mathcal{A})$ as the joint policy for the joint skill $\boldsymbol{z} \in \mathcal{Z}$ of the $\mathcal{N}$ agents, where $\mathcal{J}_{\boldsymbol{z}} = \cup_{i \in \mathcal{N}} \mathcal{J}_z^i$. Learning the safe skills of the agents becomes our low-level RL problem. Next, we detail our proposed HMARL-CBF method.

## 5.1  Multi-Agent High-Level Policy Learning

As mentioned above, the high-level policy is formulated as a joint centralized optimization problem involving all the agents over their set of skills. The optimization problem can be solved using existing CTDE schemes, either using policy gradients or value decomposition. The higher level problem can be modeled as an MSMDP and the corresponding extrinsic reward for state $\boldsymbol{s}$ and skill $\boldsymbol{z}$ executed in $k$ steps using policy $\pi_{\boldsymbol{z}}$ can be defined as: $R(\boldsymbol{s}, \boldsymbol{z}, k) = \mathbb{E}_{\boldsymbol{a}_t \sim \pi_{\boldsymbol{z}}}[\sum_{t=0}^{k-1} \gamma^t r_H(\boldsymbol{s}_t, \boldsymbol{a}_t) | \boldsymbol{s}_0 = \boldsymbol{s}]$. We call this extrinsic as it is task-specific and, hence, chosen to be the environment reward. This is revised by

incorporating safety corresponding to (6) rendering the following extrinsic reward:

$$r_H(\boldsymbol{s}, \boldsymbol{a}) = -\left[l(\boldsymbol{s}, \boldsymbol{a}) - \sum_{i \in \mathcal{N}} \sum_{j \in \mathcal{N}^i(\boldsymbol{s})} p_i^j \mathbb{I}(b_j^i(\boldsymbol{s}^i, \boldsymbol{s}^j) < 0) b(\boldsymbol{s}^i, \boldsymbol{s}^j)\right] \tag{10}$$

where $\boldsymbol{s} \in \mathcal{S}, \boldsymbol{a} \in \mathcal{A}, p_i^j \in \mathbb{R}_{>0}$ and $\mathbb{I}$ is an indicator function defined as follows.

$$\mathbb{I}(x) = \begin{cases} 1, & \text{if } x < 0 \text{ for some } i, \\ 0, & \text{otherwise.} \end{cases}$$

The minimization problem is thus mapped to a maximization problem of accumulated reward. Then, the problem in (6) can be mapped to the following problem:

$$J(\pi_H, \pi_{\boldsymbol{z}}) = \mathbb{E}_{\tau \sim (\pi_H \circ \pi_{\boldsymbol{z}})}[R_H(\tau)] = \mathbb{E}_{\tau \sim (\pi_H \circ \pi_{\boldsymbol{z}})}\left[\sum_{t'=0}^{\infty} \gamma^{k_{t'}} R(\boldsymbol{s}_{t'}, \boldsymbol{z}_{t'}, k_{t'}) | \boldsymbol{z}_{t'} \sim \pi_H\right]$$

$$= \mathbb{E}_{\boldsymbol{z}_{t'} \sim \pi_H}\left[\sum_{t'=0}^{\infty} \mathbb{E}_{\boldsymbol{s}_t, \boldsymbol{a}_t \sim \pi_{\boldsymbol{z}_{t'}}, k_{t'}}\left[\sum_{t=0}^{k_{t'}-1} \gamma^{t+k_{t'}} r_H(\boldsymbol{s}_t, \boldsymbol{a}_t)\right]\right]$$

$$\pi_H^* = \arg \max_{\pi_H \in \Pi_H} J(\pi_H, \pi_{\boldsymbol{z}}) \tag{11}$$

where $\pi_H \in \Pi_H$ is the space of the high-level policies, $\pi_{\boldsymbol{z}} \in \Pi_{\boldsymbol{z}}$ is the space of the joint skills policy of the agents and $\tau$ is a trajectory sampled from the joint composition of $\pi_H$ and $\pi_{\boldsymbol{z}}$. The policy $\pi_H^*$ given a joint skills policy of the agents $\pi_{\boldsymbol{z}}$ optimizes the trajectories with respect to the reward in (10). This problem can be solved using any existing on-policy or off-policy MARL algorithm [33, 81, 25]. If we parameterize the policy with $\theta$, the policy gradient of the parametrized policy $\pi_{H_\theta}$, as derived in [74] is given by:

$$\nabla_\theta J(\pi_{H_\theta}, \pi_{\boldsymbol{z}}) = \mathbb{E}_{\tau \sim (\pi_{H_\theta} \circ \pi_{\boldsymbol{z}})}\left[\sum_{t=0}^{\infty} \nabla_\theta \log \pi_{H_\theta}(\boldsymbol{z}_t | \boldsymbol{s}_t) \left(\sum_{t'=0}^{\infty} \gamma^{t'} R(\boldsymbol{s}_{t'}, \boldsymbol{z}_{t'}, k_{t'})\right)\right] \tag{12}$$

Under a CTDE scheme, in order to execute the policies de-centrally, the agent policies/actor network (parameters) can be learned individually agent-wise conditioned on their observation (and history $h_{t-1}^i$) i.e., $\pi_{H_\theta}^i(\cdot | \boldsymbol{o}_t^i, h_{t-1}^i)$, using a centralized critic network. Alternatively, the policy $\pi_{H_\theta}$ can be learnt by minimizing the following (Q-learning) loss using a parametrization $Q_\theta$:

$$\mathcal{L}(\theta) = \mathbb{E}_{\boldsymbol{s}_{t'}, \boldsymbol{z}_{t'}, \boldsymbol{s}_{t'+1}, k_{t'}}[(R(\boldsymbol{s}_{t'}, \boldsymbol{z}_{t'}, k_{t'}) + \gamma^{k_{t'}} \max_{\boldsymbol{z} \in \Pi_{\boldsymbol{z}}} Q_\theta^{tot}(\boldsymbol{s}_{t'+1}, \boldsymbol{z}_{t'+1}) - Q_\theta^{tot}(\boldsymbol{s}_{t'}, \boldsymbol{z}_{t'}))^2] \tag{13}$$

where $Q_\theta^{tot}$ is the total state-action value as defined in (2), which can be learned using value decomposition methods based on the Individual Global Max principle proposed in [56, 30].

## 5.2 CBF-Based Individual Safe Skill Learning

The low-level policy is associated with execution of the skills selected by the agents based on the high-level policy. We define an intrinsic step reward conditioned on any skill $z^i$ of any agent $i$ as $r_L^i(\boldsymbol{s}^i, \boldsymbol{a}^i | z^i)$ to incorporate intrinsic motivation. Notice, the safety constraints are included in the extrinsic step reward $r_H$. Additionally, to ensure safe execution of the skill, we formulate the low-level policy as a Stochastic Optimal Control Problem (SOCP) as follows:

$$\max_{\pi_{z_{t'}}^i \in \Pi_{z_{t'}}^i, k_{t'}} \mathbb{E}_{\pi_{z_{t'}}^i, \boldsymbol{s}_{t', \ldots, k_{t'}}^i, k_{t'}}\left[\sum_{t=t'}^{k_{t'}} r_L^i(\boldsymbol{s}_t^i, \boldsymbol{a}_t^i) | z_{t'}^i\right]$$

$$\text{subject to } \mathbb{E}_{\boldsymbol{s}_t}[\min\{0, b_i^j(\boldsymbol{s}_t^i, \boldsymbol{s}_t^j)\}] \geq 0 \quad \forall j \in \mathcal{N}^i(\boldsymbol{s}_t), t \in [t', \ldots k_{t'}], \boldsymbol{s}_{k_{t'}}^i \in \mathcal{J}_{z_{t'}}^i. \tag{14}$$

where $t'$ is the time the skill was initiated by agent $i$. In order to solve the problem we formulate a *parameterized* constrained optimization problem based on CBFs at each time step $t$ that gives us the deterministic policy as a function of the observation conditioned on the skill $\mu^i(\boldsymbol{o}_t^i | z_{t'}^i; \phi^i)$. We include control Lyapunov functions (CLFs) $V_i^j$ as functions of the state $\boldsymbol{s}_t^i$ and the terminal state $\boldsymbol{s}_{k_{t'}}^i \in \mathcal{J}_{z_{t'}}^i$ to assist in learning the optimal trajectory corresponding to the skill and terminate by

converging to the termination set with $T_{\max}$. This is similar to stabilizing the system at the terminal state. Note that $s_{k_t}$ need not be a terminal state, but any state-related Lyapunov function that aids in learning the skill. This results in an optimization problem with a quadratic cost function and linear constraints which makes it a Quadratic Program (QP) that requires low computation. The optimization problem corresponding to the deterministic policy $\mu^i(o_t^i|z_{t'}^i; \phi^i)$ is given below.

$$\min_{\boldsymbol{a}_t^i \in \mathcal{A}^i, \boldsymbol{e}} \quad \boldsymbol{a}_t^{i\top} H(\boldsymbol{\phi}_H^i)\boldsymbol{a}_t^i + F^\top(\boldsymbol{\phi}_F^i)\boldsymbol{a}_t^i + \boldsymbol{\phi}_e^\top \boldsymbol{e} \tag{15}$$

subject to

$$L_f^m b_i^j(\boldsymbol{s}_t^i, \boldsymbol{s}_t^j) + L_g L_f^{m-1} b_i^j(\boldsymbol{s}_t^i, \boldsymbol{s}_t^j)\boldsymbol{a}_t^i + \Omega(b_i^j(\boldsymbol{s}_t^i, \boldsymbol{s}_t^j); \boldsymbol{\phi}_b^{i,j}) + \alpha_m(\zeta_{m-1}(\boldsymbol{s}_t^i, \boldsymbol{s}_t^j); \boldsymbol{\phi}_b^{i,j}) \geq 0; \ \forall j \in \mathcal{N}^i(\boldsymbol{s}_t)$$

$$L_f V_i^j(\boldsymbol{s}_t^i, \boldsymbol{s}_{k_t}^i) + L_g V_i^j(\boldsymbol{s}_t^i, \boldsymbol{s}_{k_t}^i)\boldsymbol{a}_t^i + \eta(\boldsymbol{s}_t^i, \boldsymbol{s}_{k_t}^i; \boldsymbol{\phi}_v^{i,j}) \leq e^{i,j}; \ \forall j = 1, 2, \dots$$

where $\boldsymbol{\phi}^i = [\boldsymbol{\phi}_H^i, \boldsymbol{\phi}_F^i, \boldsymbol{\phi}_b^{i,1}, \dots, \boldsymbol{\phi}_b^{i,|\mathcal{N}^i(\boldsymbol{s}_t)|}, \boldsymbol{\phi}_v^{i,1}, \boldsymbol{\phi}_v^{i,2}, \dots]$ are the parameters of the optimization problem such that each term in the vector $\boldsymbol{\phi}^i$ belongs to $\mathbb{R}_{>0}$, $H \in \mathbb{R}^{\dim(\mathcal{A}_i) \times \dim(\mathcal{A}_i)}$, $F \in \mathbb{R}^{\dim(\mathcal{A}_i)}$ and $\boldsymbol{e}$ is the slack term from the CLF constraints. The chosen quadratic objective, combined with the affine CBF constraints, forms a Quadratic Program (QP) at each time step $t$, which is well suited for real-time safety-critical control applications.

In our case, $H$ is chosen to be positive definite as the objective generally includes tracking error/control effort minimization, which makes the problem strictly convex. The parameters $\boldsymbol{\phi}^i$ are learned as a function of the agent $i$'s observation $\boldsymbol{o}^i$ and parameterized by $\boldsymbol{\nu}^i$: $\boldsymbol{\phi}^i = \Gamma_{\boldsymbol{\nu}^i}(\boldsymbol{o}^i|z^i)$, where $\Gamma$ is differentiable w.r.t to $\boldsymbol{\nu}^i$. The function can be a deterministic or stochastic function of a continuous random variable $\epsilon$ using the reparameterization trick in [78]. With the reparameterization trick, the control policy becomes stochastic, thus allowing for use of stochastic policy gradient algorithms.

An agent's trajectory $\tau^i = \{\boldsymbol{s}_0^i, \boldsymbol{a}_0^i, \boldsymbol{s}_1^i, \boldsymbol{a}_1^i, \boldsymbol{s}_2^i, \boldsymbol{a}_2^i, \dots\} = \{\boldsymbol{s}_0^i, \boldsymbol{z}_0^i, \boldsymbol{s}_{k_0}^i, \boldsymbol{z}_{k_0}^i, \boldsymbol{s}_{k_1}^i, \boldsymbol{z}_{k_1}^i, \dots\}$ can be represented as a sequence of states and actions indexed by time as well as states and skills indexed by time on a different scale. Additionally, the forward invariant property of CBFs can guarantee that the trajectory resides in the safe set. In order to jointly learn with the high-level policy from the sampled trajectories, we map the safe skill learning problem for any agent $i$ in (14) as a trajectory optimization problem with the policy $\pi_{z_\nu}^i$ given by the following objective:

$$J^i(\pi_H, \pi_z^{-i}, \pi_{z_\nu}^i) = \mathbb{E}_{\tau^i \sim (\pi_H \circ (\pi_z^{-i}, \pi_{z_\nu}^i))}[R_L^i(\tau^i)] = \mathbb{E}\left[\sum_{t=0}^{\infty} \gamma^t r_L^i(\boldsymbol{s}_t^i, \boldsymbol{a}_t^i)|z_t^i \sim \pi_H^i, \boldsymbol{a}_t^i \sim \pi_{z_{t\nu}}^i\right] \tag{16}$$

We drop the policy parameters $\phi$ and superscript $i$ from $\boldsymbol{\nu}$ to avoid overloading the symbols. Here, $\pi_z^{-i}$ is the joint skills policy of the other agents except agent $i$. we solve the optimization problem in (16) by learning the parameters of the QP in (15) that maximizes the total reward. The problem in (16) can be solved using policy gradient (PG) methods as detailed in Appendix A.3.

**Theorem 5.1.** *The satisfaction of the CBF constraints $b_j^i$ in (15) guarantees safe execution of the skills, thus guaranteeing the safety of our proposed HMARL approach.*

The proofs for the theorems can be found in the appendix A.3. It should be noted that robust CBFs can be used in the presence of model uncertainty following [9] to achieve similar safety guarantees. Besides that, we detail how the high-level task specific reward can be incorporated to the low-level skill learning to align the low-level learning to the task.

An example of a skill in autonomous driving context is cruising. This skill causes the vehicle to move at a constant speed for a fixed distance. We choose the reward as the error norm of the velocity. And, a velocity CLF constraint in (36) is incorporated in the skill QP based policy by setting $v_{des} = v_{current}$ where $v_{current}$ is the current velocity of the vehicle. And the termination constraint is chosen as the target distance.

## 6 Experiments

### 6.1 Environment Details

We validate our proposed method on traffic scenarios involving complex road networks with multiple agents that must ensure safety concerning neighboring agents while navigating toward their destinations. We use the METADRIVE simulator [35] to implement, validate, and compare our method

against existing baselines in a cooperative setting, where agents (autonomous vehicles) collaborate to improve the overall traffic efficiency.

The five environments from the METADRIVE simulator are illustrated in Figure 2. We implement our algorithm in four of the originally proposed environments and design a new environment to simulate an on-ramp merging roadway, replacing the parking lot environment. Since safety is less challenging in parking lots due to low vehicle speeds, the problem can be formulated and solved as a static optimization problem with a fixed number of vehicles and parking spaces. These environments involve regions where agents can cross path with other agents. Hence, it is necessary for the agents to cooperate but at the same time stay safe with each other to achieve the task i.e., reach the destination successfully and timely.

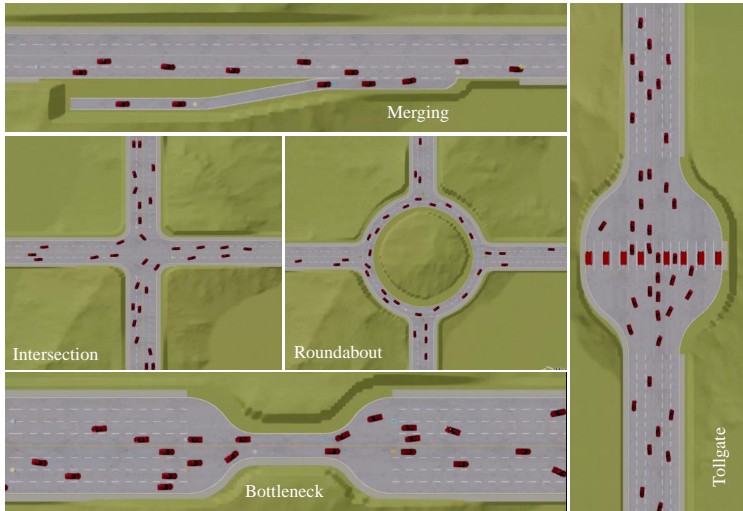

Figure 2: Illustration of the various METADRIVE environments.

## 6.2 Implementation Details

We have implemented our proposed method using both on-policy and off-policy algorithms. Specifically, we implement our method based on a joint and group-based policy learning approach which are detailed in section A.4. The on-policy implementation was done using MAPPO [81] with PPO [60] for the skills. The off-policy implementation used QMIX [56] for the higher level policy and SAC [26] for learning the agent skills. The extrinsic reward was chosen to be same as that defined in METADRIVE (which includes the energy, time, driving reward, lane deviation, collision penalty) of all the AVs in the environment. Besides that we include some regularization terms corresponding to our skills to avoid overfitting to any specific skill(s). The algorithms were implemented using RLLib [75]; an open-source framework for ML. Further details about the implementation can be found in A.1.

## 6.3 Results and Baseline Comparison

We compare our algorithms against the state-of-the-art baselines in METADRIVE environment namely, CoPo [49], the independent policy optimization (IPPO) method [14] using PPO [60] as the individual learners, and Mean Field MARL with centralized critic [80] and form the mean field policy optimization (MFPO). The curriculum learning (CL) [45] is also a baseline, where the training was divided into 4 phases and gradually the number of initial agents in each episode from 25% to 100% of the target number. We ran our algorithm and the benchmark algorithms for 5 seeds over 300k and 1M iterations respectively. It is important to mention that the benchmark algorithms use 2000+ hours of individual driving experience in addition to the training iterations to achieve the illustrated results. *On the other hand, we jointly optimize both the RL policies from same data, thus not introducing any additional sample complexity due to the hierarchal decomposition.* The training results include the success rate and average episode length to illustrate the efficacy and validate our proposed approach.

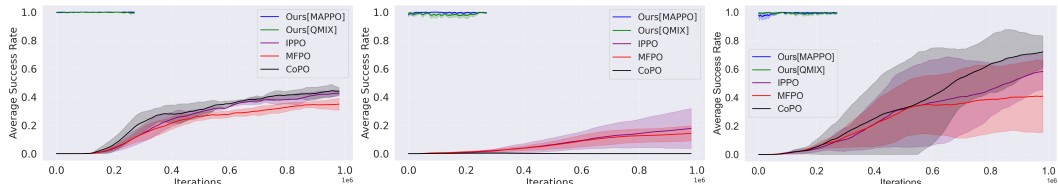

Figure 3: From left to right: success rate in merging, tollgate, and bottleneck environments.

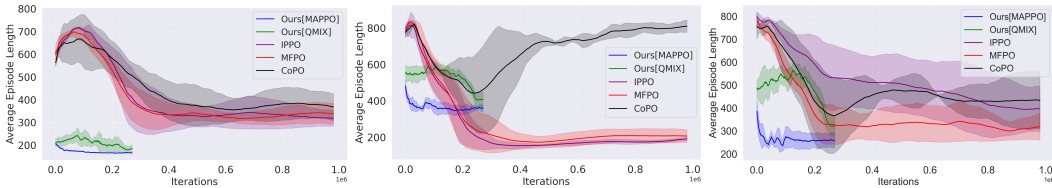

Figure 4: From left to right: average episode length in merging, tollgate, and bottleneck environments.

Notably, the efficiency metric defined by the authors of CoPo [49]—calculated as the inverse of the average episode length divided by the difference between the number of successful and unsuccessful agents—further supports the validity of our chosen metrics. We present the results of training for the Merging, Bottleneck, and Tollgate environments in Figures 3 and 4, respectively. Additional environment results can be found in appendix A.1.

The key observation is that our algorithm achieves almost 100 % success rate with an approximate model of the vehicle (see also appendix A.1.1) from the beginning of training, while also optimizing the joint policy evident from the average episode length. Although several baselines achieve better average episode length in the tollgate environment, the success rate is very low, implying most agents crash out, resulting in the remaining agents finishing faster, returning a lower average episode length. With the hierarchical decomposition, our algorithms converge faster ($\leq$ 300k) compared to the baselines which we run for 1M iterations, thus confirming the claim about expediting convergence.

Table 1: Summary of results for Merging, Intersection, Roundabout, Bottleneck, and Tollgate Environments. $\uparrow$: higher is better; $\downarrow$: lower is better. **Bold:** the best performance for each metric.

| Environments | Metrics | MFPO | CoPo | MAPPO - Lagrangian | MACPO | HMARL-CBF (on-policy) | HMARL-CBF (on-policy w/ Grouping) | HMARL-CBF (off-policy) | HMARL-CBF (off-policy w/ Grouping) |
|---|---|---|---|---|---|---|---|---|---|
| Merging | Success $\uparrow$ | $0.43_{\pm0.09}$ | $0.48_{\pm0.01}$ | $0.34_{\pm0.12}$ | $0.32_{\pm0.04}$ | $\mathbf{0.99_{\pm0.01}}$ | $\mathbf{0.99_{\pm0.00}}$ | $\mathbf{0.99_{\pm0.01}}$ | $\mathbf{0.99_{\pm0.01}}$ |
| | Time $\downarrow$ | $762.32_{\pm47.97}$ | $669.46_{\pm72.08}$ | $588.23_{\pm58.82}$ | $531.25_{\pm31.68}$ | $\mathbf{164.52_{\pm2.64}}$ | $187.91_{\pm64.65}$ | $169.60_{\pm7.61}$ | $166.70_{\pm9.26}$ |
| | Energy $\downarrow$ | $14.44_{\pm2.51}$ | $13.81_{\pm1.25}$ | $29.41_{\pm2.94}$ | $33.96_{\pm6.44}$ | $14.10_{\pm0.23}$ | $\mathbf{12.14_{\pm1.83}}$ | $14.73_{\pm0.76}$ | $13.92_{\pm0.09}$ |
| Intersection | Success $\uparrow$ | $0.59_{\pm0.16}$ | $0.73_{\pm0.09}$ | $0.08_{\pm0.03}$ | $0.17_{\pm0.03}$ | $0.97_{\pm0.04}$ | $0.93_{\pm0.05}$ | $\mathbf{0.96_{\pm0.03}}$ | $\mathbf{0.96_{\pm0.01}}$ |
| | Time $\downarrow$ | $725.39_{\pm222.51}$ | $687.69_{\pm141.90}$ | $2250.39_{\pm513.42}$ | $1058.22_{\pm117.64}$ | $317.41_{\pm33.92}$ | $388.27_{\pm64.92}$ | $\mathbf{264.28_{\pm32.90}}$ | $280.11_{\pm32.00}$ |
| | Energy $\downarrow$ | $6.52_{\pm0.27}$ | $5.68_{\pm0.39}$ | $21.25_{\pm4.33}$ | $28.23_{\pm0.13}$ | $5.56_{\pm0.04}$ | $\mathbf{3.62_{\pm0.68}}$ | $5.78_{\pm0.24}$ | $5.11_{\pm0.20}$ |
| Roundabout | Success $\uparrow$ | $0.85_{\pm0.04}$ | $0.80_{\pm0.03}$ | $0.02_{\pm0.00}$ | $0.23_{\pm0.02}$ | $\mathbf{0.99_{\pm0.01}}$ | $\mathbf{0.99_{\pm0.04}}$ | $0.98_{\pm0.03}$ | $0.98_{\pm0.02}$ |
| | Time $\downarrow$ | $404.00_{\pm17.88}$ | $515.05_{\pm18.83}$ | $12500.00_{\pm2600.92}$ | $1304.34_{\pm86.76}$ | $403.18_{\pm21.66}$ | $375.67_{\pm29.31}$ | $370.21_{\pm67.04}$ | $\mathbf{326.26_{\pm24.13}}$ |
| | Energy $\downarrow$ | $12.07_{\pm0.67}$ | $12.31_{\pm0.26}$ | $104_{\pm25.55}$ | $43.78_{\pm2.69}$ | $9.89_{\pm0.23}$ | $\mathbf{7.75_{\pm0.39}}$ | $10.15_{\pm0.82}$ | $11.41_{\pm0.69}$ |
| Bottleneck | Success $\uparrow$ | $0.48_{\pm0.16}$ | $0.75_{\pm0.07}$ | $0.06_{\pm0.02}$ | $0.16_{\pm0.01}$ | $\mathbf{0.99_{\pm0.00}}$ | $\mathbf{0.99_{\pm0.02}}$ | $\mathbf{0.99_{\pm0.01}}$ | $\mathbf{0.99_{\pm0.01}}$ |
| | Time $\downarrow$ | $679.83_{\pm158.33}$ | $651.54_{\pm63.52}$ | $1667.77_{\pm355.96}$ | $1336.88_{\pm143.66}$ | $241.16_{\pm8.33}$ | $262.94_{\pm59.10}$ | $313.80_{\pm22.00}$ | $\mathbf{240.47_{\pm28.52}}$ |
| | Energy $\downarrow$ | $9.95_{\pm1.39}$ | $6.93_{\pm0.61}$ | $66.67_{\pm16.29}$ | $40.7_{\pm0.61}$ | $7.37_{\pm0.00}$ | $\mathbf{4.86_{\pm0.68}}$ | $7.29_{\pm0.08}$ | $7.49_{\pm0.03}$ |
| Tollgate | Success $\uparrow$ | $0.11_{\pm0.04}$ | $0.00_{\pm0.00}$ | $0.00_{\pm0.00}$ | $0.00_{\pm0.00}$ | $\mathbf{0.99_{\pm0.02}}$ | $0.98_{\pm0.02}$ | $0.98_{\pm0.01}$ | $\mathbf{0.99_{\pm0.01}}$ |
| | Time $\downarrow$ | $1884.36_{\pm319.82}$ | - | - | - | $369.32_{\pm5.82}$ | $427.02_{\pm32.43}$ | $377.42_{\pm9.99}$ | $\mathbf{307.82_{\pm8.09}}$ |
| | Energy $\downarrow$ | $53.63_{\pm7.72}$ | - | - | - | $8.40_{\pm0.05}$ | $\mathbf{5.17_{\pm0.44}}$ | $8.93_{\pm0.11}$ | $8.99_{\pm0.01}$ |

Next, we evaluate the efficacy of our algorithm in optimizing the task objective against the baseline algorithms. Specifically, we compare against MFPO and COPO, as well as MAPPO-Lagrangian and the MACPO algorithm introduced in [24]. We use the success-weighted metrics for evaluation and comparison with the baseline, ablation, and generalization experiments. The reason for weighting the metrics by success rate is that poorly performing algorithms with very low success rate (agents crashing and getting removed from the episode rapidly) can achieve superior average energy and time, which is debunked through this. Besides *success rate*, we use *success weighted average energy* and *success weighted average time* which are the average energy and episode length of all agents during an episode divided by the success rate.

The evaluation results are presented in Table 1. As can be seen, our method achieves almost 100% success rate which is significantly better than the baselines including MAPPO-Lagrangian and MACPO which are based on the cMDP framework. Also, we achieve better time and energy efficiency compared to the baselines across all environments. This goes to highlight that our methods learns a better cooperative policy in addition to enhancing the safety of the agents.

### 6.4 Ablation Results

In this subsection, we present ablation studies to further highlight the merit of our proposed approach using the HMARL-CBF on-policy implementation. The results are presented in Table 2. First, we remove the hierarchical structure to implement a flat MAPPO policy using various penalties on the safety constraints. However, a flat policy fails to learn in these environments. Next, we replace the cooperative high-level policy with a uniform random selection of skills across agents. This causes a reduction in success rates as well as performance across all environments. Subsequently, we investigate the necessity of pointwise-in-time constraints via random constraints dropout—masking road boundary constraints in the low-level policy with a Bernoulli distributed random variable (success Probability 0.5)—which uniformly reduces success rates and exhibits the largest performance decline in environments where agents cross paths across several zones in the environment (i.e. safety-critical). Next, we substitute the low-level skill with an parameterized Linear Quadratic Regulator (LQR)-based (removing the CBFs) within the hierarchical MARL framework. The LQR based skill solves a fixed final state control problem without incorporating safety constraints. This causes a significant decline in success rate as well as in success-weighted time and energy efficiency. Finally, we examine fixed low-level policy parameterization by selecting parameters at the boundaries of the feasible space without prior knowledge, which further diminishes performance across all metrics across the environments.

Table 2: Summary of ablation results for all the environments (Pf is penalty factor).

| Environments | Metrics | Flat Policy (Pf = 1) | Flat Policy (Pf = 5) | Flat Policy (Pf = 10) | Uniform high-level | Random Constraints Dropout | Hierarchical MARL with LQR | Fixed low-level parameterization | HMARL-CBF |
|---|---|---|---|---|---|---|---|---|---|
| Merging | Success ↑ | 0 | 0 | 0 | $0.94_{\pm0.09}$ | $0.91_{\pm0.04}$ | 0 | $0.52_{\pm0.06}$ | $\mathbf{0.99_{\pm0.01}}$ |
| | Energy ↓ | - | - | - | $14.17_{\pm0.49}$ | $15.30_{\pm0.31}$ | - | $22.30_{\pm0.12}$ | $\mathbf{14.10_{\pm0.23}}$ |
| | Time ↓ | - | - | - | $211.70_{\pm23.80}$ | $183.91_{\pm1.49}$ | - | - | $\mathbf{164.52_{\pm2.64}}$ |
| Intersection | Success ↑ | 0 | 0 | 0 | $0.86_{\pm0.04}$ | $0.80_{\pm0.04}$ | $0.14_{\pm0.02}$ | 0 | $\mathbf{0.97_{\pm0.04}}$ |
| | Energy ↓ | - | - | - | $5.75_{\pm0.08}$ | $5.00_{\pm0.11}$ | $50.00_{\pm0.71}$ | - | $\mathbf{5.56_{\pm0.04}}$ |
| | Time ↓ | - | - | - | $574.30_{\pm60.60}$ | $320.50_{\pm68.17}$ | $921.42_{\pm40.71}$ | - | $\mathbf{317.41_{\pm33.92}}$ |
| Roundabout | Success ↑ | 0 | 0 | 0 | $0.68_{\pm0.08}$ | $0.60_{\pm0.08}$ | 0 | 0 | $\mathbf{0.99_{\pm0.01}}$ |
| | Energy ↓ | - | - | - | $13.23_{\pm1.08}$ | $8.50_{\pm0.74}$ | - | - | $\mathbf{9.89_{\pm0.23}}$ |
| | Time ↓ | - | - | - | $907.35_{\pm50.98}$ | $600.03_{\pm34.67}$ | - | - | $\mathbf{403.18_{\pm21.66}}$ |
| Bottleneck | Success ↑ | 0 | 0 | 0 | $0.65_{\pm0.03}$ | $0.55_{\pm0.09}$ | 0 | 0 | $\mathbf{0.99_{\pm0.00}}$ |
| | Energy ↓ | - | - | - | $10.03_{\pm0.10}$ | $12.03_{\pm0.04}$ | - | - | $\mathbf{7.37_{\pm0.00}}$ |
| | Time ↓ | - | - | - | $923.07_{\pm61.73}$ | $965.45_{\pm24.18}$ | - | - | $\mathbf{241.16_{\pm8.33}}$ |
| Tollgate | Success ↑ | 0 | 0 | 0 | $0.91_{\pm0.03}$ | $0.92_{\pm0.03}$ | 0 | 0 | $\mathbf{0.99_{\pm0.02}}$ |
| | Energy ↓ | - | - | - | $8.68_{\pm0.03}$ | $9.35_{\pm0.02}$ | - | - | $\mathbf{8.40_{\pm0.05}}$ |
| | Time ↓ | - | - | - | $713.00_{\pm24.15}$ | $423.40_{\pm15.60}$ | - | - | $\mathbf{369.32_{\pm5.82}}$ |

We also perform generalization experiments to assess how well our policies to various environmental changes. The results for the experiments can be found in appendix A.1.3.

## 7 Conclusion

In conclusion, we addressed the problem of learning safe policies in multi-agent, safety-critical autonomous systems with pointwise-in-time safety constraints. We proposed HMARL-CBF, a novel hierarchical MARL approach leveraging Control Barrier Functions (CBFs). Our method employs a skill-based hierarchy, learning a high-level policy for selecting skills and a low-level policy for safely executing them. We validated our approach in complex multi-agent environments, demonstrating its effectiveness through extensive empirical results. Future work can explore jointly learning both the set of skills and their corresponding policies, along with the cooperative behavior.

### Acknowledgments

The authors thank the anonymous reviewers for their invaluable feedback and constructive suggestions. This work was supported in part by NSF under grants CCF-2340776, ECCS-1931600, DMS-1664644, CNS-2149511, and by ARPA-E under grant DE-AR0001282.

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

# A Appendix / supplemental material

## A.1 Experimental Details and Additional Results

We begin this section by presenting the details of the training, and evaluation experiments. Besides that, we include results for the generalization experiments used to validate our proposed method.

In addition to the METADRIVE environment, we evaluated our method on a suite of lidar-based environments introduced in [85], as illustrated in Figure 5. All environments are partially observable, with agents equipped with lidar sensors. In the *Target* environments, each agent is assigned a fixed goal and must reach it while avoiding collisions with other agents and obstacles. The primary distinction between the *Target* and *Target-Bicycle* environments lies in the underlying agent dynamics; further details are provided in Figure 5. In the *Spread* environment, agents must cooperate to cover all goal locations while avoiding both obstacles and one another.

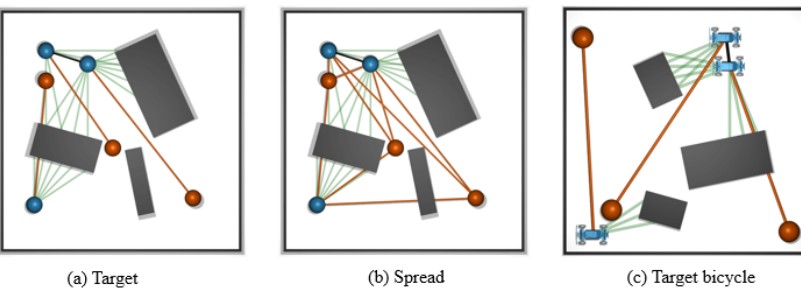

(a) Target         (b) Spread         (c) Target bicycle

Figure 5: Illustration of the lidar based environments which include: (a) target environment, (b) spread environment and (c) target bicycle environment.

### A.1.1 Training Setup and Results.

**METADRIVE Simulator.** In all five environments used in our experiments, the vehicles are initialized at random spawn points, and assigned destinations to each of them randomly. The agents are terminated in three situations: reaching the destinations (deemed as successful), crashing with others or driving out of the road (deemed as failure). Once an existing vehicle is terminated new vehicles are spawned immediately to ensure continuous traffic flow. The terminated vehicles remains static for 10 steps (2s) in the original positions, creating impermeable obstacles. If a vehicle crashes into other terminated vehicles before it is removed from the environment, it is also considered as failure. The *success rate* is the fraction of the successful agents to the total number of spawned agents. Thus, the environment allows for the total number of vehicles to vary in time and exceed the the initial number of agents which mimics real-world challenging traffic scenarios.

We trained with 15 vehicles for merging, 30 vehicles for intersection and roundabout, 25 vehicles for bottleneck and 40 vehicles for tollgate respectively. As mentioned previously, the extrinsic reward corresponding to (10) was set to the environment reward, besides the regularization terms. The intrinsic reward is described in subsection A.4. We present the training results for the remaining three environments from figure 4 i.e. merging, roundabout and intersection in figure 6.

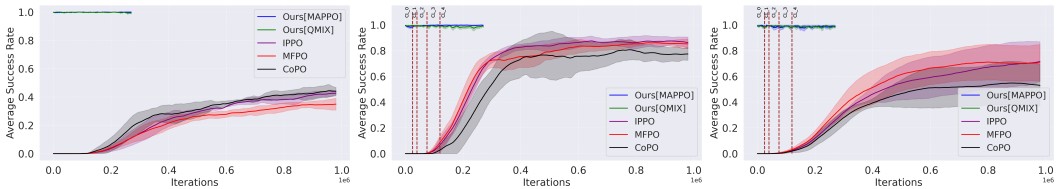

Figure 6: Plot of the success rate of the agents with our algorithm vs the baselines in merging, roundabout, and intersection environments respectively. CL refers to curriculum learning, four different curriculum were used with varying traffic densities. The benchmark methods use 2000+ hours of individual driving data prior to the training.

As can be noticed, our algorithm achieves a near perfect safety rate with both on-policy and off-policy algorithms during training, and, keeping consistent with the tollgate and bottleneck environments are trained for 300k iterations. However, in order to encourage exploration in roundabout and intersection environments for learning the cooperative behavior, we use curriculum learning by varying the arrival rates, while keeping the traffic density constant. The curriculum transitions happened at 30k, 50k, 80k and 125k iterations during training.

### A.1.2 Evaluation Setup and Results

**METADRIVE Simulator.** As mentioned previously we used *success rate weighted average time and energy* for evaluating the performance of our algorithms. In order to compare against the baseline methods, for each environment and algorithm (e.g. on-policy, on-policy w/Grouping, off-policy and so on) we train using five seeds. Then we run five episodes for each seed and generate the evaluation results by computing the mean and standard deviation over the 25 runs. In addition to the presented results in Table 2, we run the baseline algorithms by changing the penalty factors for the safety related terms (both inter-agent and environment safety) in the environment reward for the bottleneck environment. The results are presented in Table 3. The primary takeaway is that the added emphasis on safety does not render better results for the baseline algorithms compared to our method. Besides that, different algorithms perform differently with an increase in penalty factor, thus not exhibiting any trend between penalty and success rate.

Table 3: Summary of baseline comparison using various penalty factors for Bottleneck environment. ↑: higher is better; ↓: lower is better. **Bold:** the best performance for each metric.

| Penalty Factors | Metrics | IPPO | CL | MFPO | CoPo | HMARL-CBF (on-policy) |
|---|---|---|---|---|---|---|
| 1 | Success ↑ | $0.09_{\pm0.03}$ | $0.34_{\pm0.15}$ | $0.48_{\pm0.16}$ | $0.75_{\pm0.07}$ | $\mathbf{0.99_{\pm0.00}}$ |
| | Time ↓ | $2644.44_{\pm353.44}$ | $939.35_{\pm147.21}$ | $679.83_{\pm158.33}$ | $651.54_{\pm63.52}$ | $\mathbf{241.16_{\pm8.33}}$ |
| | Energy ↓ | $69.66_{\pm9.77}$ | $11.61_{\pm2.02}$ | $9.95_{\pm1.39}$ | $6.93_{\pm0.61}$ | $\mathbf{7.37_{\pm0.00}}$ |
| 5 | Success ↑ | $0.58_{\pm0.11}$ | $0.395_{\pm0.01}$ | $0.615_{\pm0.08}$ | $0.716_{\pm0.02}$ | $\mathbf{0.99_{\pm0.00}}$ |
| | Time ↓ | $568_{\pm56}$ | $646.5_{\pm43.16}$ | $713_{\pm273.7}$ | $529.13_{\pm79}$ | $\mathbf{241.16_{\pm8.33}}$ |
| | Energy ↓ | $30_{\pm41.5}$ | $10.8_{\pm0.24}$ | $8.64_{\pm0.65}$ | $7.69_{\pm0.02}$ | $\mathbf{7.37_{\pm0.00}}$ |
| 10 | Success ↑ | $0.18_{\pm0.08}$ | $0.55_{\pm0.11}$ | $0.42_{\pm0.23}$ | $0.655_{\pm0.04}$ | $\mathbf{0.99_{\pm0.00}}$ |
| | Time ↓ | $1292.17_{\pm647}$ | $606.49_{\pm204.89}$ | $1414_{\pm973}$ | $514_{\pm126.92}$ | $\mathbf{241.16_{\pm8.33}}$ |
| | Energy ↓ | $40.97_{\pm27}$ | $9.14_{\pm0.93}$ | $22.12_{\pm22.08}$ | $8.37_{\pm0.04}$ | $\mathbf{7.37_{\pm0.00}}$ |

**Lidar Environment Suite.**

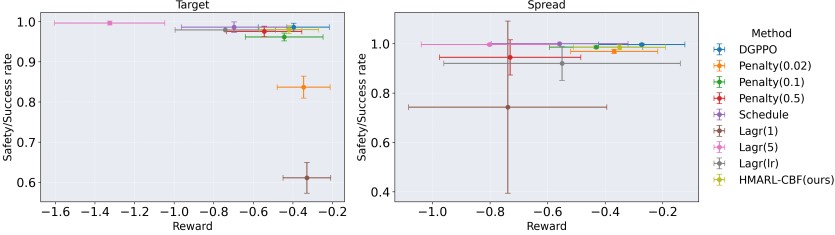

Figure 7: Success/safety rate vs reward for Target and Spread environment.

**Baselines.** We compare our method with DGPPO [85], InforMARL [46] and MAPPO-Lagrangian [24], three state-of-the-art methods for multi-agent safe control. For InforMARL, we evaluate fixed and scheduled penalty variants. Further details about the baselines can be found in [85]. For MAPPO-Lagrangian, we test two settings for the Lagrange multiplier and one learning rate. Performance is better if the plot is located in top right corner. As can be seen, our method achieves a high success rate while also achieving high reward compared to the baselines in [85]. Finally, we present the results for the bicycle target environment in Figure 8 which shows that our method scales with number of agents.

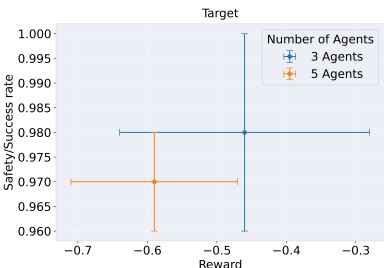

Figure 8: Success/safety rate vs reward for Bicycle Target environment.

### A.1.3  Generalization Results

Besides the evaluation experiments, we conducted a series of experiments to assess the generalizability of our proposed approach. Firstly we systematically vary the lane widths in the environments. The corresponding success rates are reported in Table 4. Our analysis indicates that reducing lane width has a minimal impact on success rates in the roundabout and bottleneck environments. However, a more comparatively greater decline is observed in the intersection environment. This performance degradation is primarily attributed to the default intersection configuration (Figure 1), where two lanes are available in each direction. Agents are trained to execute lane changes from both lanes depending on their assigned routes. When lane width is reduced, agents attempting to perform lane changing from the inside lane at times gets stuck (and, eventually get eliminated from the episode due to timeout), or hit the road boundary which causes the drop in success rate. Conversely, increasing lane width generally maintains or enhances success rates across all environments.

Secondly, we investigated skill transferability by training low-level skills in an intersection environment and subsequently using them to train a high-level policy across multiple environments. Namely, we investigate the transferability in the merging and roundabout environments. This is because the observation is different and distinct for the Bottleneck and Tollgate environments compared to the other environments. The results are presented in Table 6. This analysis reveals that skills learned in one environment can generalize effectively to others producing comparatively similar performance. In the roundabout environment, we observe a minor decline in the success rate and increase in the weighted average energy and time; yet our approach outperforms the baselines presented in Table 2.

Finally, we test the generalizability of our method for varying traffic density/ number of agents. The results for this test are presented in Tables 7 and 8 respectively. For every environment we trained for the default density as shown in the table and tested for lower and higher values. From the results, we can conclude that the success rate is unaffected by traffic density. The average time of the agents increases with increasing traffic density as expected. Finally, the energy values show a decreasing trend with increase in traffic density across most environments. This was anticipated because energy has a negative correlation with time.

### A.1.4  Computer Details.

We conducted our experiments on a desktop equipped with a single NVIDIA RTX A5000 GPU and 32 CPU cores. Given that our environments involve a high number of agents (up to 45 agents in the tollgate environment), the primary computational cost is attributed to data collection. Training and convergence are achieved efficiently, as demonstrated in the training and evaluation results. Furthermore, we utilize shared neural network architectures across agents, which significantly reduces the overall computational cost and training time.

### A.2  Background

**Definition A.1** (Relative degree). The relative degree of a (sufficiently many times) differentiable function $b : \mathbb{R}^n \to \mathbb{R}$ with respect to system (3) is the number of times it needs to be differentiated along its dynamics until the control $a$ explicitly appears in the corresponding derivative.

Table 4: Success rate across different environments for varying lane widths.

| Lane Width (m) | Success Rate | | | | |
|---|---|---|---|---|---|
| | **Merging** | **Intersection** | **Roundabout** | **Bottleneck** | **Tollgate** |
| 3.25 | $0.99_{\pm 0.01}$ | $0.90_{\pm 0.03}$ | $0.98_{\pm 0.02}$ | $0.97_{\pm 0.02}$ | $0.99_{\pm 0.001}$ |
| 3.5 (default) | $0.99_{\pm 0.01}$ | $0.97_{\pm 0.04}$ | $0.99_{\pm 0.01}$ | $0.99_{\pm 0.003}$ | $0.99_{\pm 0.02}$ |
| 4 | $0.99_{\pm 0.01}$ | $0.97_{\pm 0.015}$ | $0.99_{\pm 0.01}$ | $0.99_{\pm 0.001}$ | $0.99_{\pm 0.02}$ |

Table 6: Summary of skill transfer results (* indicates runs done by transferring skills).

| Experiments | Success | Energy | Time |
|---|---|---|---|
| Merging | $0.99_{\pm 0.01}$ | $14.24_{\pm 0.23}$ | $166.18_{\pm 2.64}$ |
| Merging* | $0.96_{\pm 0.016}$ | $14.49_{\pm 0.58}$ | $178.34_{\pm 5.90}$ |
| Roundabout | $0.99_{\pm 0.01}$ | $9.89_{\pm 0.23}$ | $403.18_{\pm 21.66}$ |
| Roundabout* | $0.96_{\pm 0.016}$ | $11.39_{\pm 0.08}$ | $404.16_{\pm 18.6}$ |

Table 7: Summary of results for varying number of agents in Merging, Intersection, and Roundabout environments.

| Metric | Merging | | | Intersection | | | Roundabout | | |
|---|---|---|---|---|---|---|---|---|---|
| | 10 | 15 (default) | 20 | 20 | 30 (default) | 40 | 20 | 30 (default) | 40 |
| Success | $0.997_{\pm 0.0}$ | $0.99_{\pm 0.01}$ | $0.98_{\pm 0.02}$ | $0.955_{\pm 0.02}$ | $0.97_{\pm 0.04}$ | $0.94_{\pm 0.04}$ | $0.99_{\pm 0.01}$ | $0.99_{\pm 0.0}$ | $0.98_{\pm 0.0}$ |
| Energy | $14.03_{\pm 0.36}$ | $14.24_{\pm 0.23}$ | $14.47_{\pm 0.18}$ | $5.85_{\pm 0.07}$ | $5.82_{\pm 0.04}$ | $5.94_{\pm 0.05}$ | $10.48$ | $10.32$ | $9.95$ |
| Time | $165.18_{\pm 2.15}$ | $166.18_{\pm 2.64}$ | $169.55_{\pm 2.15}$ | $318.54_{\pm 19}$ | $332.3_{\pm 36}$ | $361.8_{\pm 20.46}$ | $389.34$ | $404.38$ | $415.03$ |

Table 8: Summary of results for varying number of agents in the Bottleneck and Tollgate environments.

| Metric | Bottleneck | | | Tollgate | | |
|---|---|---|---|---|---|---|
| | 20 | 25 (default) | 30 | 30 | 40 (default) | 45 |
| Success | $0.99_{\pm 0.01}$ | $0.99_{\pm 0.003}$ | $0.978_{\pm 0.02}$ | $0.99_{\pm 0.02}$ | $0.99_{\pm 0.02}$ | $0.99_{\pm 0.01}$ |
| Energy | $7.47_{\pm 0.03}$ | $7.44_{\pm 0.001}$ | $7.43_{\pm 0.02}$ | $8.51_{\pm 0.02}$ | $8.51_{\pm 0.048}$ | $8.50_{\pm 0.02}$ |
| Time | $227.22_{\pm 7.77}$ | $243.59_{\pm 8.33}$ | $244.62_{\pm 6.62}$ | $371.53_{\pm 6.78}$ | $371.05_{\pm 5.82}$ | $374.02_{\pm 8.48}$ |

For a constraint $b(s) \geq 0$ with relative degree $m$, $b : \mathbb{R}^n \to \mathbb{R}$, and $\zeta_0(s) := b(s)$, we define a sequence of functions $\zeta_i : \mathbb{R}^n \to \mathbb{R}, i \in \{1, \ldots, m\}$ :

$$\zeta_i(s) := \dot{\zeta}_{i-1}(s) + \alpha_i\left(\zeta_{i-1}(s)\right), \ i \in \{1, \ldots, m\}, \tag{17}$$

where $\alpha_i(), \ i \in \{1, \ldots, m\}$ denotes a $(m-i)^{\text{th}}$ order differentiable class $\mathcal{K}$ function. We further define a sequence of sets $C_i, \ i \in \{1, ..., m\}$ associated with (17) which take the following form,

$$C_i := \{s \in \mathbb{R}^n : \zeta_{i-1}(s) \geq 0\}, \ i \in \{1, ..., m\}. \tag{18}$$

**Definition A.2** (High Order CBF (HOCBF) [76, 77])**.** Let $C_1, ..., C_m$ be defined by (18) and $\zeta_1(s), ..., \zeta_m(s)$ be defined by (17). A function $b : \mathbb{R}^n \to \mathbb{R}$ is a High Order Control Barrier Function (HOCBF) of relative degree $m$ for system (3) if there exists $(m-i)^{th}$ order differentiable class $\mathcal{K}$ functions $\alpha_i, \ i \in \{1, ..., m-1\}$ and a class $\mathcal{K}$ function $\alpha_m$ such that

$$\sup_{a \in \mathcal{A}} [L_f^m b(s) + L_g L_f^{m-1} b(s)a + \Omega(b(s)) + \alpha_m(\zeta_{m-1}(s))] \geq 0 \tag{19}$$

for all $s \in \bigcap_{i=1}^m C_i$. In (19), $L_f^m$ and $L_g$ denotes derivative along $f$ and $g$ $m$ times and one time respectively, and $S(.)$ denotes the remaining Lie derivative along $f$ with degree less than or equal to $m-1$ (omitted for simplicity, see [76]).

Note that the HOCBF in (19) is a general form of the degree one CBF [3] ($m = 1$) and exponential CBF in [47]. The following theorem on HOCBFs implies the forward invariance property of the

CBFs and the original safety set. The proof is omitted (see [76] for the proof). If the dynamics are of the form: $\dot{s} = f(s, a)$, we can express it in the form:

$$\dot{s}' = f'(s', a) + g'(s')a' \tag{20}$$

where $s' = \begin{bmatrix} s \\ a \end{bmatrix}$ and $a'$ are the augmented system states and new primitive actions of the system respectively, $f' = \begin{bmatrix} f(s, a) \\ \mathbf{0}_{\dim(a)} \end{bmatrix}$, and $g' = \begin{bmatrix} \mathbf{0}_{\dim(s) \times \dim(a')} \\ B \end{bmatrix}$, and $B \in \mathbb{R}^{\dim(a) \times \dim(a')}$. Generally, $B$ contains ones in its diagonal entries. The redefined dynamics can be used subsequently for defining CBFs.

### A.3 Safe Skill learning

We provide the details about the computation of policy gradients necessary to learn the skill policy parameters in this subsection. We follow [4] by applying the Karush-Kuhn-Tucker condition on the Lagrangian to derive the gradient of the state feedback policy with respect to its parameters. Specifically, let $\lambda$ denote the dual variables on the HOCBF and CLF constraints, let $D(\cdot)$ create a diagonal matrix from a vector, and let $\mu^*, \lambda^*$ denote the optimal solutions of $\mu$ and $\lambda$, respectively. We can then write $d_\mu$ and $d_\lambda$ in the form:

$$\begin{bmatrix} d_\mu \\ d_\lambda \end{bmatrix} = \begin{bmatrix} H & G^T D(\lambda^*) \\ G & D(Ga^* - h) \end{bmatrix}^{-1} \begin{bmatrix} \mathbf{1}_{\dim(\mathcal{A}_i)} \\ 0 \end{bmatrix} \tag{21}$$

where $G, h$ are concatenated by $G_b^j, G_v^j, h_b^j, h_v^j$,

$$G_b^j = -L_g L_f^{m-1} b_i^j(s^i, s^j); \quad j \in \mathcal{N}^i(s)$$
$$G_v^j = -L_g V_i^j(s^i, s^j; \phi_v^{i,j}); \quad j = 1, 2, \dots$$
$$h_b^j = L_f^m b_i^j(s^i, s^j) + \Omega\left(b_j^i(s^i, s^j); \phi_b^{i,j}\right) + \alpha_m\left(\zeta_{m-1}\left(s^i, s^j; \phi_b^{i,j}\right)\right); j \in \mathcal{N}^i(s)$$
$$h_b^j = L_f V_i^j(s^i, s^j) + \eta(s^i, s^j; \phi_v^{i,j}); \quad j = 1, 2, \dots \tag{22}$$

As the control bounds in (15) are not trainable, they are not included in $G$ and $h$. Then, the relevant gradients with respect to all the parameters are given by

$$\nabla_H \mu = \frac{1}{2}\left(d_u u^T + u d_u^T\right), \nabla_F \mu = d_u \tag{23}$$

$$\nabla_G \mu = D(\lambda^*)\left(d_\lambda u^T + \lambda d_u^T\right), \nabla_h \mu = -D(\lambda^*) d_\lambda \tag{24}$$

With the hierarchical composition of the policies $(\pi_H \circ \pi_z)$, we define the augmented state $\hat{s} = [s^i, s^{-i}, z, s_{z,init}]$ where $s^{-i}$ is the state of the agents other than agent $i$ and $s_{z,init}$ is the joint state of the agents at the time of the initiation of the joint skill $z$. The initiation state is included as the joint termination of the skill depends on the initiation state. The state and action value functions for agent $i$ corresponding to reward $r_L^i$, skill policy $\pi_{z^i}$ (given the high level policy $\pi_H$ and the skill policy other agents $\pi_{z^{-i}}$) are defined as follows:

$$V^{\pi_{z^i}}(\hat{s}) = \mathbb{E}[(r_L^i(\hat{s}_0, a_0^i) + \mathbb{E}_{\hat{s}_1}[Q(\hat{s}_1, a_1^i)])|\hat{s}_0 = \hat{s}]$$
$$= \mathbb{E}[(r_L^i(s_0^i, a_0^i) + \mathbb{E}_{\hat{s}_1}[Q(\hat{s}_1, a_1^i)])|s_0^i = s^i, z^i] \tag{25}$$

The state-action value function for agent $i$, given the hierarchical policy composition $\pi_H \circ \pi_z$, is defined as:

$$Q^{\pi_{z^i}}(\hat{s}, a^i) = \mathbb{E}\left[r_L^i(\hat{s}, a^i) + \gamma \mathbb{E}_{\hat{s}'}\left[V^{\pi_{z^i}}(\hat{s}')\right]\right] \tag{26}$$

The corresponding advantage function is given by:

$$A^{\pi_{z^i}}(\hat{s}, a^i) = Q^{\pi_{z^i}}(\hat{s}, a^i) - V^{\pi_{z^i}}(\hat{s}) \tag{27}$$

Based on the above definition of value functions and advantage function, as well the gradient of QP controller output with respect to its parameters, standard on-policy, and off-policy algorithms can be used to derive the policy gradient to learn the controller parameters.

**Proof of Theorem 5.1**

*Proof.* Given, the constraints $b_i^j$'s are satisfied at the initial time, the satisfaction of the CBF constraints makes the corresponding safety sets forward invariant per Theorem (3.4), thus guaranteeing their satisfaction at all future times. This makes the policies corresponding to the skills of the agents safe, thus, guaranteeing the safety of our proposed HMARL approach. $\qquad\square$

**High-level task specific reward to low level learning.** The low-level policies should be such that they optimize the entire trajectory given by the problem in (6). Hence, in addition to introducing intrinsic motivation through the low-level reward, inspired by [36], we introduce an advantage term to the low-level reward resulting in the following revised low-level reward:

$$\tilde{r}_L^i(\boldsymbol{s}_t^i, \boldsymbol{a}_t^i | z_{t'}^i) = \lambda \frac{A^{\pi_H}(\boldsymbol{s}_{t'}, \boldsymbol{z}_{t'})}{\mathcal{N}} + (1-\lambda) r_L^i(\boldsymbol{s}_t^i, \boldsymbol{a}_t^i | z_{t'}^i) \tag{28}$$

where $\lambda \in [0,1]$ and $t' \le t$ is the time when the joint skill at time $t$ was initiated. The high-level advantage $A^{\pi_H}(\boldsymbol{s}_{t'}, \boldsymbol{a}_{t'})$ is defined as below:

$$A^{\pi_H}(\boldsymbol{s}_{t'}, \boldsymbol{z}_{t'}) = Q^{\pi_H}(\boldsymbol{s}_{t'}, \boldsymbol{z}_{t'}) - V^{\pi_H}(\boldsymbol{s}_{t'}) \tag{29}$$

where $Q^{\pi_H}(\boldsymbol{s}_{t'}\boldsymbol{z}_{t'})$ and $V^{\pi_H}(\boldsymbol{s}_{t'})$ are as defined in (2) and (1) respectively. The parameter $\lambda$ can be used to balance between skill learning and optimization of the joint behavior of the agents. From [36] we have that for high level policies the following holds:

$$J(\tilde{\pi}_H) = J(\pi_H) + \mathbb{E}_{\boldsymbol{s}_{t'}, \boldsymbol{z}_{t'} \sim \pi_H} \left[ \sum_{t'=0} \gamma^{t'} A^{\pi_H}(\boldsymbol{s}_{t'}, \boldsymbol{z}_{t'}) \right] \tag{30}$$

**Proposition A.3.** *Given the low level reward in* (28)*, the revised low level learning objective $\tilde{J}^i$ for agent $i$ can be expressed as:*

$$\tilde{J}^i(\pi_H, \pi_{\boldsymbol{z}^{-i}}, \pi_{z^i}) \approx \frac{\lambda}{\mathcal{N}(1-\gamma)} \mathbb{E} \left[ \sum_{t'=0} \gamma^{t'} A^{\pi_H}(\boldsymbol{s}_{t'}, \boldsymbol{z}_{t'}) \right] + (1-\lambda) J^i(\pi_H, \pi_{\boldsymbol{z}^{-i}}, \pi_{z^i}) \tag{31}$$

*Proof.* Based on 28, we can write our low level policy's learning objective for agent $i$ as the following:

$$\tilde{J}^i(\pi_H, \pi_{\boldsymbol{z}^{-i}}, \pi_{z^i}) = \mathbb{E} \left[ \sum_{t'=0} \mathbb{E}_{\boldsymbol{s}_t^i, \boldsymbol{a}_t^i \sim \pi_{z_{t'}^i}} \left[ \sum_{t=0}^{k_{t'}-1} \gamma^{t+k_{t'}} \tilde{r}_L^i(\boldsymbol{s}_t^i, \boldsymbol{a}_t^i | z_{t'}^i) \right] \right]$$

$$= \mathbb{E} \left[ \sum_{t'=0} \mathbb{E}_{\boldsymbol{s}_t^i, \boldsymbol{a}_t^i \sim \pi_{z_{t'}^i}} \left[ \sum_{t=0}^{k_{t'}-1} \gamma^{t+k_{t'}} \left( \lambda \frac{A^{\pi_H}(\boldsymbol{s}_{t'}, \boldsymbol{z}_{t'})}{\mathcal{N}} + (1-\lambda) \tilde{r}_L^i(\boldsymbol{s}_t^i, \boldsymbol{a}_t^i | z_{t'}^i) \right) \right] \right]$$

$$= \frac{\lambda}{\mathcal{N}} \mathbb{E} \left[ \sum_{t'=0} \frac{\gamma^{t'}(1-\gamma^{k_{t'}})}{1-\gamma} A^{\pi_H}(\boldsymbol{s}_{t'}, \boldsymbol{z}_{t'}) \right] + (1-\lambda) \mathbb{E} \left[ \sum_{t'=0} \mathbb{E}_{\boldsymbol{s}_t^i, \boldsymbol{a}_t^i \sim \pi_{z_{t'}^i}} \left[ \sum_{t=0}^{k_{t'}-1} \gamma^{t+k_{t'}} r_{L_A}^i(\boldsymbol{s}_t^i, \boldsymbol{a}_t^i | z_{t'}^i) \right] \right]$$

$$\approx \frac{\lambda}{\mathcal{N}(1-\gamma)} \mathbb{E} \left[ \sum_{t'=0} \gamma^{t'} A^{\pi_H}(\boldsymbol{s}_{t'}, \boldsymbol{z}_{t'}) \right] + (1-\lambda) J^i(\pi_H, \pi_{\boldsymbol{z}^{-i}}, \pi_{z^i})$$

$$\qquad\qquad\qquad\qquad\qquad\qquad\qquad\qquad\qquad\qquad\qquad\qquad\qquad\qquad\qquad\qquad\qquad\qquad\qquad\qquad\square$$

Similarly, the low-level reward for every agent can be revised as above to align the skill learning process with the task learning process, as the first term is related to the high-level policy return.

### A.4 Algorithm Implementation

#### A.4.1 Algorithm details

As mentioned previously, our method can be implemented using both existing on-policy and off-policy RL algorithms. The bilevel RL problems presented in (11) and (16), respectively, should be solved sequentially. However, as previous works illustrated [34], jointly optimizing both policies by updating their corresponding parameters alternatively renders similar performances, which is consistent with our observation from the experiments. Based on this observation, the algorithm is presented in (1). The high-level policy $\pi_{H_\theta}$ is decomposed into agent-level parameterized policies $\pi_{H_{\theta^i}}^i$ to learn decentralized policies.

---

**Algorithm 1** Hierarchical Multi-Agent Reinforcement Learning with CBFs

---

1: Initialize randomly the high-level policy parameters $\boldsymbol{\theta}$ and parameters $\boldsymbol{\nu}$ of the low-level policy parameters $\boldsymbol{\phi}$           //network parameter sharing done between agents.
2: **for** each episode **do**
3:      $z_0^i \sim \pi_{H_{\boldsymbol{\theta}}}(z^i|\boldsymbol{o}_0^i), \; \forall i \in \mathcal{N}$               // sample initial skill
4:      Set data buffer $\mathcal{D} = \{\}$
5:      **for** $t \leftarrow 0, \ldots, T$ **do**
6:          Get action for agent $i$ based on the selected skill $a_t^i = \mu^i(\boldsymbol{o}_t^i|z_{t'}^i; \boldsymbol{\phi}_{\boldsymbol{\nu}}) \; \forall i \in \mathcal{N}$     // sample primitive action
7:          Get $s_{t+1}$ and $r_H$ from environment and $r_L^i \; \forall i \in \mathcal{N}$
8:          Sample skill for agent $i$, $z_{t+1}^i \sim \pi_{H_{\boldsymbol{\theta}}}^i(z^i|\boldsymbol{o}_t^i), \; \forall i \in \mathcal{N}_z(t)$    // $\mathcal{N}_z(t)$ is the list of agents who have to update their skills based on termination scheme used.
9:      **end for**
10:     Update high level policy network $\boldsymbol{\theta}$ according to (12), or, (13), using sampled data.
11:     Update low level policy network parameters $\boldsymbol{\nu}$ based on (23) and (24), using sampled data
12: **end for**

---

We implement the algorithm using two different approaches inspired by existing works in the area of MARL. Notably, we implemented the algorithm in the following ways. It is important to note that under all these approaches, the hierarchical policy is learned agent-wise to allow for decentralized execution.

1. Joint policy learning: This approach involves solving (11) by optimizing the high-level policies of the agents jointly by learning the policy parameters $\boldsymbol{\theta}^i$s.

2. Group policy learning: This approach is inspired by the idea of learning the value function by decomposing the joint value function into the value function of groups of agents. In our fully cooperative setting, the total step reward can be composed as a sum of the rewards of the individual agents, i.e. $r_H(\boldsymbol{s}, \boldsymbol{a}) = \sum_{i=1}^N r_H^i(\boldsymbol{s}^i, \boldsymbol{a}^i)$. Let $\{G^j\}_{j=1}^{N_G}$ represent the groups with each $G^j \subset \mathcal{N}$ such that, $G^j \cap G^{j'} = \varnothing$ and $\mathcal{N} = \cup_{i=1}^{N_G} G^i$. We can define the group reward as the sum over the individual reward of the agents in the group. Then, under this approach we can express the objective in (11) as below (We only include the necessary symbols in the expectation to avoid symbol overload):

$$
\begin{aligned}
J(\pi_H, \pi_{\boldsymbol{z}}) &= \mathbb{E}\left[\sum_{t'=0}^{\infty} \mathbb{E}_{\boldsymbol{a}_t \sim \pi_{\boldsymbol{z}_{t'}}}\left[\sum_{t=0}^{k_{t'}-1} \gamma^{t+k_{t'}} r_H(\boldsymbol{s}_t, \boldsymbol{a}_t)\right]\right] \\
&= \mathbb{E}\left[\sum_{t'=0}^{\infty} \mathbb{E}_{\boldsymbol{a}_t \sim \pi_{\boldsymbol{z}_{t'}}}\left[\sum_{t=0}^{k_{t'}-1} \gamma^{t+k_{t'}} \sum_{j=1}^{N_G} \underbrace{\sum_{i=1}^{N} r_H^i(\boldsymbol{s}_t^i, \boldsymbol{a}_t^i)\mathbb{I}[i \in G^j]}_{=r_H^{G^j}(\boldsymbol{s}_t^{G^j}, \boldsymbol{a}_t^{G^j})}\right]\right] \\
&= \sum_{j=1}^{N_G} \mathbb{E}\left[\sum_{t'=0}^{\infty} \mathbb{E}_{\boldsymbol{a}_t^{G^j} \sim \pi_{\boldsymbol{z}_{t'}}}\left[\sum_{t=0}^{k_{t'}-1} r_H^{G^j}(\boldsymbol{s}_t^{G^j}, \boldsymbol{a}_t^{G^j})\right]\right]
\end{aligned}
\tag{32}
$$

Here, $\boldsymbol{s}^{G^j}$ and $\boldsymbol{a}^{G^j}$ are the states and actions of the agents in group $G^j$, The agent skill policies are learnt independently of each other. Hence. the joint skills policies can be written as $\pi_{\boldsymbol{z}} = \prod_{i=1}^{N} \pi_{z^i} = \prod_{j=1}^{N_G} \prod_{i \in G^j} \pi_{z^i} = \prod_{j=1}^{N_G} \pi_{\boldsymbol{z}^{G^j}}$ where $\pi_{\boldsymbol{z}^{G^j}}$ is the joint skills policy of the agents in the group $G^j$. Then, (32) can be written as:

$$
\begin{aligned}
J(\pi_H, \pi_{\boldsymbol{z}}) &= \sum_{j=1}^{N_G} \mathbb{E}\left[\sum_{t'=0}^{\infty} \mathbb{E}_{\boldsymbol{a}_t^{G^j} \sim \pi_{\boldsymbol{z}_{t'}^{G^j}}}\left[\sum_{t=0}^{k_{t'}-1} r_H^{G^j}(\boldsymbol{s}_t^{G^j}, \boldsymbol{a}_t^{G^j})\right]\right] \\
&= J^1(\pi_H^{G^1}) + \cdots + J^{N_G}(\pi_H^{G^{N_G}})
\end{aligned}
$$

where, $k_{t'}$ now becomes the time at which any agent in the group changes selects/switches to a new skill. We drop the low-level policy corresponding to skills for brevity. This approach

enables better scalability as the overall problem is decomposed into one that learns policies for the groups; however, this comes at the expense of performance as the original optimization problem involves all the agents.

Note that all these implementations retain the safety guarantee as it is offered by our safe skill-learning approach.

### A.4.2 Implementation Details: METADRIVE Simulator

Metadrive environments illustrated in figure 2 by default, include 3 basic sensors: lidar, sideDetector and laneLineDetector which are used for detecting moving objects, sidewalks/solid lines, and broken/solid lines respectively. The lidar state observation returns a state vector containing necessary information for navigation tasks. The details of the observation can be found here [2]. The environmental safety constraints include i. the road boundaries and ii. the booth for tollgate environment only. And, each agent has to stay safe to all neighboring agents which are within the lidar sensing radius. It is important to mention that we only use a dynamic bicycle model for the CBF constraints, the environment uses pybullet physics engine for simulation. The kinematic bicycle model of the vehicle is provided in (33).

$$
\begin{aligned}
\dot{p} &= \frac{v \cos(\psi)}{1 - d\kappa(p)}, \\
\dot{d} &= v \sin(\psi), \\
\dot{\psi} &= \frac{v}{L} \tan(\delta) - \kappa(p)\dot{p}, \\
\dot{v} &= a.
\end{aligned}
\tag{33}
$$

where $p$ represents the position of the vehicle along the reference path, $d$ is the lateral deviation of the vehicle from the reference path, $\psi$ denotes the yaw angle of the vehicle (heading angle relative to the path), and $v$ is the longitudinal velocity of the vehicle. The control inputs are $\delta$, which is the steering angle, and $a$, which is the acceleration (or deceleration). The parameter $L$ denotes the wheelbase of the vehicle, and $\kappa(p)$ is the curvature of the reference path at position $s$. It is important to add that, these states are provided by the environment for each ego vehicle/agent in its observation.

In order to incorporate safety with respect to other agents and environmental obstacles, we define simple circles-based constraints as illustrated in Figure 9. And, for the road boundaries we use the lateral deviation from the center lane to define safety functions. Let $p^j$ and $d^j$ denote the distance along the path and lateral deviation of another agent $j$ with respect to agent $i$, located within the sensing radius of agent $i$. This information is provided in the observation of agent $i$ i.e. $(d^i, p^j, d^j) \in \mathcal{O}^i$. Note that, for an ego vehicle $p^i$ is always 0 in its body frame. let $\Delta(p^i, d^i, p^j, d^j)$ denote the euclidean distance between agent $i$ and $j$ respectively. Similarly, let $p^e$ and $d^e$ denote the longitudinal and lateral distance of an environmental obstacle, like a tollgate booth, as illustrated in Figure 9 from agent $i$, and $\Delta(p^i, d^i, p^e, d^e)$ be the euclidean distance. Then, the safety constraints corresponding to other agent $j$, environmental constraints for an agent $i$ are given below:

$$
\Delta(p^i, d^i, p^j, d^j)^2 \geq r(v^i) \,, \Delta(p^i, d^i, p^e, d^e)^2 \geq e; \,, \text{and } |d^i| \leq c
\tag{34}
$$

where $r(v^i)$ is a monotonically increasing function of $v^i$, $e \in \mathbb{R}_{>0}$ and $c \in \mathbb{R}_{>0}$.

Consider the CBF for the constraint for inter-agent safety defined as $h(\mathcal{O}^i) = \Delta(p^i, d^i, p^j, d^j) - r(v^i)$ where $\Delta(p^i, d^i, p^j, d^j) = \sqrt{(p^i - p^j)^2 + (d^i - d^j)^2}$, then the corresponding CBF constraint is given by:

$$
\underbrace{2(p^i - p^j)\left(\frac{v^i \cos(\psi^i)}{1 - d^i\kappa(p^i)} - \dot{p}^j\right) + 2(d^i - d^j)\left(v^i \sin(\psi^i) - \dot{d}^j\right)}_{L_f h(\mathcal{O}^i)} + \underbrace{-2r(v^i)\frac{dr(v^i)}{dv^i}a^i}_{L_g h(\mathcal{O}^i)a^i}
$$
$$
+ \underbrace{\alpha\left((p^i - p^j)^2 + (d^i - d^j)^2 - r(v^i)^2\right)}_{h(\mathcal{O}^i)} \geq 0
\tag{35}
$$

---

[2] https://metadrive-simulator.readthedocs.io/en/latest/obs.html

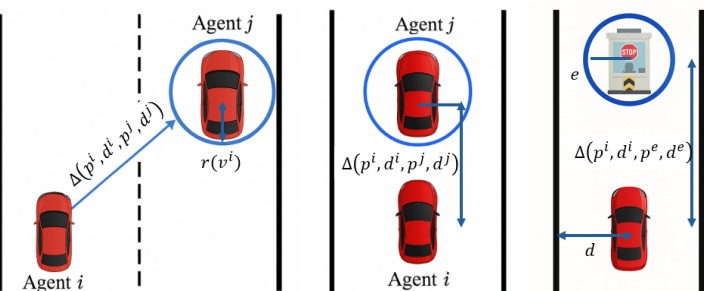

Figure 9: From left to right: illustration of i. inter agent safety constraints on another lane, ii. inter agent safety constraints on same lane and ii. environmental constraints.

Additionally, we define CLFs for state reference tracking namely velocity and heading reference tracking. As we define termination state set corresponding to each skill, we define CLFs to drive the system to the termination state to be able to successfully execute the skill. However, it is important to note that the CLFs are not defined specific to skills and, hence, are generalized to the task. The CLFs corresponding to velocity and heading reference are given by the equation below. where $v_t^i$ and $v_{t_f}^i$ are the velocities at We parameterize the corresponding CLF constraint which in case of the velocity and heading constraints becomes the following:

$$V(v^i, v_{des}^i) = (v^i - v_{des}^i)^2; \qquad (36)$$
$$V(\psi^i, \psi_{des}^i) = (\psi^i - \psi_{des}^i)^2 \qquad (37)$$

where $v_{des}$ is the desired velocity and $\psi_{des}$ is the desired heading of the vehicle.

**Skills description.** We define five skills for each agent for the METADRIVE environments which are: i. cruising, ii. accelerating (speed up), iii. yielding (slowing down), iv. left lane changing, and v. right lane changing.

*Cruising.* As the name suggests, this skill causes the vehicle to move at a constant speed for a fixed distance. The velocity CLF constraint in (36) is incorporated in the skill QP based policy by setting $v_{des} = v_{current}$ where $v_{current}$ is the current velocity of the vehicle.

*Accelerating/speeding up* This skill corresponds to an agent increasing its longitudinal velocity. Let the initial velocity at the time of the initiation of the skill be $v_0$. The skill terminates when the velocity reaches $v_T = v_0 + dv$, where $dv \in \mathbb{R}_{>0}$ denotes the desired velocity increment and $v_T$ is the velocity of the agent at the time of termination. Hence, the $v_{des}$ is set to $v_0 + dv$ for this skill in the low level QP based skill policy. This skill facilitates timely task completion and helps avoid congestion by preventing agents from unnecessarily slowing down the traffic flow.

*Yielding/slowing down. Yielding.* This skill enables an agent to reduce its longitudinal velocity. Let $v_0$ be the velocity at skill initiation. The skill terminates when the velocity decreases to $v_T = v_0 - dv$, where $dv \in \mathbb{R}_{>0}$ is the desired reduction and $v_T$ is the velocity at the time of termination of the skill. Thus, the velocity CLF in (36) is incorporated to the QP based skill policy by setting $v_{des} = v_0 - dv$ Yielding allows agents to safely navigate dense traffic and cooperate by yielding to agents approaching from other directions, thereby facilitating efficient joint task completion.

*Left Lane Changing.* This skill enables an agent to switch to the adjacent left lane. Let $d_{left,0}$ denote the distance to the left road boundary and $d_0$ be the lateral deviation from the center lane at the initiation of the skill. Let $d_{lane}$ denote the lane width, provided by the environment. The target lateral position is computed as $d_{left,T} = d_{left,0} - d_{lane} + d_0$. We define the desired heading angle as a function of $e$ i.e. $\phi_{des} = f(e)$, where $e = d_{left,T} - d_{left,0}$, and incorporate it into the CLF constraint on heading in (37) via a QP-based low-level skill control policy. This skill is essential for scenarios such as tollgates, bottlenecks, and merging zones, where adaptive lateral movement is required to maintain flow and avoid congestion.

*Right Lane Changing.* This skill enables an agent to switch to the adjacent right lane. Let $d_{right,0}$ denote the distance to the right road boundary and $d_0$ be the lateral deviation from the center lane at the initiation of the skill. Let $d_{lane}$ denote the lane width, provided by the environment. The target

lateral position is computed as $d_{right,T} = d_{right,0} - d_{lane} + d_0$. We define the desired heading angle as a function of $e$, i.e., $\phi_{des} = f(e)$, where $e = d_{right,T} - d_{right,0}$, and incorporate it into the CLF constraint on heading in (37) via a QP-based low-level skill control policy. This skill is useful in scenarios such as avoiding obstacles, preparing for highway exits, or facilitating merging from the right, enabling smoother and safer lane coordination.

As mentioned previously we include a timeout to ensure finite time termination of the skills for algorithmic sanity.

**Intrinsic reward** The intrinsic reward serves as an internal shaping signal that guides each agent toward desirable driving behaviors—smoothness, lane-centering, and reference-tracking—independently of any external task reward. By combining several negative penalties, the agent is discouraged from abrupt maneuvers, drifting off its lane, or deviating from its target speed and heading.

$$ r_{L}^{i} = -c_1 \|\boldsymbol{a}^i\|^2 - c_2 \left( \frac{v^i - v_{des}}{v_{des}} \right)^2 - c_3 \left( \frac{\psi^i - \psi_{des}}{\pi} \right)^2 - c_4 \|d^i\|^2 \tag{38} $$

Where $c_1, c_2, c_3, c_4, c_5$ are the coefficients of each terms. The intrinsic reward combines penalties on aggressive acceleration and steering, deviations from reference speed and heading, and lateral offset from the lane center, with a small forward-progress bonus. Together, these terms encourage smooth, centered, and efficient driving without abrupt maneuvers.

**CBF-Based Safe Skill Implementation.** As described earlier, we parameterize the QP-based deterministic policy $\mu(\boldsymbol{o} \mid z, \boldsymbol{\phi})$ for each skill $z$, as detailed in Section 5.2. The QP formulation is parameterized through the objective, as well as the CBF and CLF constraints. In particular, we introduce parameters for the class-$\mathcal{K}$ function in the CBF constraint, the shape and size of the circular safety regions, and the convergence rate of the CLF constraint. To ensure validity, all parameters are constrained to be nonnegative using box constraints of the form $(0, \phi_H]$, where $\phi_H$ is a fixed upper bound applied uniformly across all parameters.

### A.4.3 Implementation Details: Lidar Suite Simulator

**Target.** Agents dynamics in this environment are governed by double integrator dynamics with continuous-time states $\boldsymbol{x}_i = [p_x^i, p_y^i, v_x^i, v_y^i]^\top \in \mathbb{R}^4$, where $\boldsymbol{p}_i = [p_x^i, p_y^i]^\top \in \mathbb{R}^2$ denotes position and $\boldsymbol{v}_i = [v_x^i, v_y^i]^\top \in \mathbb{R}^2$ denotes velocity; control inputs $\boldsymbol{u}_i = [a_x^i, a_y^i]^\top \in \mathbb{R}^2$ represent accelerations along the Cartesian axes, and the agent dynamics follow $\dot{\boldsymbol{x}}_i = [v_x^i, v_y^i, a_x^i, a_y^i]^\top$, with velocity and control constraints bounded in $[-10, 10]$ and $[-1, 1]$ respectively.

**Spread.** The underlying dynamics, state representation, control parametrization, and bounds are identical to those in the Target environment, i.e., agents evolve under double integrator dynamics with four-dimensional state vectors comprising positions and velocities, and two-dimensional control inputs that modulate accelerations; the bounded constraints on velocities and controls likewise remain $[-10, 10]$ and $[-1, 1]$, respectively.

**Target Bicycle** In the Target Bicycle environment, agents dynamics are modeled using a bicycle model where the state vector $\boldsymbol{x}_i = [p_x^i, p_y^i, \cos\theta^i, \sin\theta^i, v^i]^\top \in \mathbb{R}^5$ encapsulates the agent's position, heading orientation via its sine and cosine, and scalar speed; the control input $\boldsymbol{u}_i = [\delta^i, a^i]^\top$ includes the steering angle $\delta^i$ and linear acceleration $a^i$, with dynamics specified as $\dot{\boldsymbol{x}}_i = [v\cos\theta, v\sin\theta, -v\sin\theta\tan\delta, v\cos\theta\tan\delta, a]^\top$, where $v \in [-10, 10]$ and $\delta \in [-1.47, 1.47]$ are the admissible bounds on speed and steering angle.

Similar to the METADRIVE environments, we define CBFs based on distance/proximity as in (34). In our formulation, we do not employ a separate heading CLF; instead, we regulate agent behavior using only the velocity CLF by aligning the desired velocity vector with the desired heading direction. Specifically, given a desired heading direction $\theta_{des}$, we compute a target velocity vector as $\boldsymbol{v}_{des} = v_{mag}[\cos(\theta_{des}), \sin(\theta_{des})]^\top$, where $v_{mag}$ is a task-specific magnitude (e.g., maximum safe speed or adaptive goal-directed speed). This $\boldsymbol{v}_{des}$ is then used in the velocity CLF described below to guide both the speed and heading behavior of the agent.

For the Target and Spread environments, where the agent's longitudinal and lateral velocities $v_x^i, v_y^i$ are explicit components of the state, we define the Control Lyapunov Function (CLF) to track a

desired velocity vector $\boldsymbol{v}_{\text{des}} = [v_{x,\text{des}}, v_{y,\text{des}}]^\top \in \mathbb{R}^2$ as:

$$V(v_x^i, v_y^i) = \frac{1}{2}\left((v_x^i - v_{x,\text{des}})^2 + (v_y^i - v_{y,\text{des}})^2\right).$$

Given the double integrator dynamics, where the control inputs $a_x^i$ and $a_y^i$ directly affect the velocity via $\dot{v}_x^i = a_x^i$ and $\dot{v}_y^i = a_y^i$, the time derivative of the CLF becomes:

$$\dot{V} = (v_x^i - v_{x,\text{des}})a_x^i + (v_y^i - v_{y,\text{des}})a_y^i.$$

To enforce convergence, we include the following CLF constraint in the QP:

$$(v_x^i - v_{x,\text{des}})a_x^i + (v_y^i - v_{y,\text{des}})a_y^i \leq -\alpha\left((v_x^i - v_{x,\text{des}})^2 + (v_y^i - v_{y,\text{des}})^2\right),$$

where $\alpha > 0$ is a tunable rate parameter.

In the Target Bicycle environment, each agent's velocity is represented as a scalar speed $v^i \in \mathbb{R}$, and the longitudinal acceleration $a^i \in \mathbb{R}$ serves as the control input directly affecting the speed via $\dot{v}^i = a^i$. To drive $v^i$ to a desired speed $v_{\text{des}} \in \mathbb{R}$, we define the CLF as:

$$V(v^i) = \frac{1}{2}(v^i - v_{\text{des}})^2.$$

The time derivative of this CLF, using $\dot{v}^i = a^i$, is:

$$\dot{V} = (v^i - v_{\text{des}})a^i.$$

To enforce convergence of the velocity to the target, the following CLF constraint is imposed in the QP:

$$(v^i - v_{\text{des}})a^i \leq -\alpha(v^i - v_{\text{des}})^2,$$

where $\alpha > 0$ is a tunable parameter determining the exponential rate of convergence.

**Skills Description.** We define four interpretable low-level skills for each agent in the LiDAR suite environments, which are: i. speeding up, ii. slowing down, iii. turning left, and iv. turning right.

*Speeding Up.* This skill allows an agent to increase its translational velocity to navigate more efficiently through open space or move rapidly towards its assigned goal. Let the velocity at the start of the skill be $\boldsymbol{v}_0 \in \mathbb{R}^2$; the skill is considered complete once the velocity reaches $\boldsymbol{v}_T = \boldsymbol{v}_0 + \Delta\boldsymbol{v}$, where $\Delta\boldsymbol{v} \in \mathbb{R}^2$, with $\|\Delta\boldsymbol{v}\| > 0$. The CLF constraint associated with velocity tracking is enforced by setting the desired velocity $\boldsymbol{v}_{\text{des}} = \boldsymbol{v}_0 + \Delta\boldsymbol{v}$ in the QP-based low-level control policy. This skill facilitates faster progression in low-density scenarios while maintaining safety guarantees via CBFs.

*Slowing Down.* This skill enables the agent to reduce its speed when approaching dynamic obstacles, other agents, or goal regions where precise navigation is required. Let the initial velocity be $\boldsymbol{v}_0 \in \mathbb{R}^2$, and define the skill termination condition as reaching $\boldsymbol{v}_T = \boldsymbol{v}_0 - \Delta\boldsymbol{v}$, where $\Delta\boldsymbol{v} \in \mathbb{R}^2$ and $\|\Delta\boldsymbol{v}\| > 0$. To enforce this behavior, the velocity CLF is specified using $\boldsymbol{v}_{\text{des}} = \boldsymbol{v}_0 - \Delta\boldsymbol{v}$. This skill is critical for smooth and safe deceleration, reducing the risk of collisions and improving maneuverability near goals.

*Turning Left.* This skill enables the agent to redirect its heading counterclockwise to avoid obstacles or navigate toward an alternate subgoal. Let the initial heading be $\theta_0$, and the desired heading be $\theta_{\text{des}} = \theta_0 + \Delta\theta$, where $\Delta\theta > 0$. Instead of using a heading CLF, we convert the desired heading into a velocity vector using a fixed target magnitude $v_{\text{mag}}$, and set $\boldsymbol{v}_{\text{des}} = v_{\text{mag}}[\cos(\theta_{\text{des}}), \sin(\theta_{\text{des}})]^\top$. This desired velocity is then tracked via the standard velocity CLF in the QP. The turning-left skill is useful for curvature adjustment in cluttered or multi-agent scenes.

*Turning Right.* This skill enables clockwise heading correction using the same velocity CLF formulation. Given initial heading $\theta_0$ and desired adjustment $\Delta\theta > 0$, the target heading is $\theta_{\text{des}} = \theta_0 - \Delta\theta$, which is converted to a velocity vector as $\boldsymbol{v}_{\text{des}} = v_{\text{mag}}[\cos(\theta_{\text{des}}), \sin(\theta_{\text{des}})]^\top$. The agent's translational motion is then regulated by the velocity CLF, enforcing convergence to the desired heading direction through velocity tracking. This skill is particularly effective in obstacle-rich or conflicting zones where sharp directional changes are required.

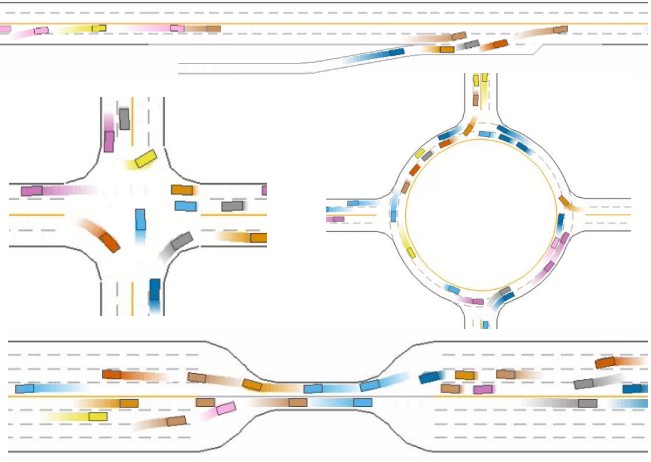

Figure 10: Visualization of agents' trajectories under the trained model: for each vehicle, its ten most recent positions are shown in a unique hue, with brightness fading from light (earlier points) to dark (later points) to indicate temporal progression.

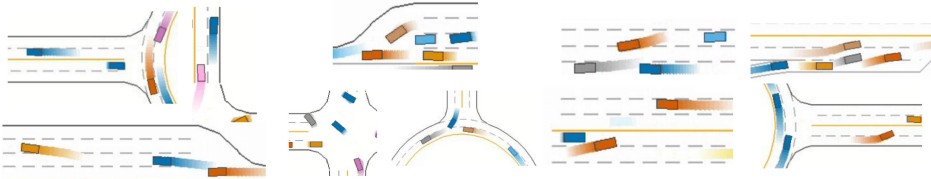

Figure 11: Visualization of skills under the trained model. From left to right: (a) acceleration (speed-up), (b) yielding (slow-down), and (c) lane changing

## A.5 Additional Results

Here, we present results to further demonstrate the merit of our proposed algorithm. Specifically, we examine the trajectories in detail to analyze the distinct skills exhibited by the agents, offering insight into how our method sustains an extremely high success rate while also outperforming the baselines in terms of energy efficiency and episode duration. To this end, we employ a visualization scheme in which each agent's ten most recent positions are depicted in a distinct color, with brightness gradually diminishing from light (earlier points) to dark (later points) to convey temporal progression. An example of this visualization is depicted in Figure 10. By looking more closely, we can identify some of the skills that the agents exhibit during the episode. For instance, as agents approach their destinations (i.e., exit points of the environment), they execute speed up skill only when safety constraints are satisfied, thereby enhancing road throughput and reducing average travel time. Several such examples are shown in Figure 11a. Yielding (i.e., controlled slow down/deceleration) is another common skill agents employ to maintain a safe distance from other vehicles. By anticipating the need to yield, agents avoid abrupt braking, which both enhances safety and reduces overall episode energy consumption. Several such examples are shown in Figure 11b. Another skill exhibited by agents is lane changing: when executed safely at appropriate locations, it enhances road throughput and consequently reduces the average episode duration. An example of such lane changing is depicted in 11c where the agent with brown color is changing its lane— creating a safe gap into which the grey agent then merges onto the main road.

