# OpenReview forum: "HMARL-CBF – Hierarchical Multi-Agent Reinforcement Learning with Control Barrier Functions for Safety-Critical Autonomous Systems"
_NeurIPS.cc/2025/Conference — NeurIPS 2025 poster_

### Official Review · Reviewer_MQj4 · 2025-06-16

**Clarity:** 2
**Significance:** 3
**Originality:** 3
**Rating:** 5
**Confidence:** 3

**Summary:**

This paper focuses on the safety of multi-agent reinforcement learning (MARL), while it emphasizes that agents have to meet safety requirements at all times. Based on Control Barrier Functions, it proposes a hierarchical approach where the higher-level problem involves learning a joint policy over the skills and lower-level problem involves learning policies to execute skills. In complex environments with a large number of agents, the method manages to present good performance in experiments.

**Questions:**

1.As is mentioned in previous works (such as HATRPO and MAFOCOPS), the agents will affect with each other. Although the proposed method in this paper can deal with multiple agents, it seems to mainly consider each single agent. I’m curious whether this will lead to some instability during the training.
2.As the method relies on the CBF and hierarchical skill learning, which I think both require some prior knowledge. From the paper, I think the authors are very familiar with the experiment environments. But if a new or very complex environment is encountered, how will the method perform?
3.In the METADRIVE experiments, the chosen baselines do not seem to be algorithms related for safety constraints. How will other methods concerned with safety perform in the environment?

**Ethical Concerns:**

["NO or VERY MINOR ethics concerns only"]

**Final Justification:**

The response to my review solves the majority of my questions. Considering the outstanding performance of the algorithm in environments with large amount of agents, I would like to raise my score.

**Limitations:**

The authors claim to discuss the limitations in both the conclusion and appendix sections but I don't find the discussion. I think the authors should analysis the computation complexity as the proposed method is based on a hierarchical framework, which tends to be more complex. Also, how the method performs if a new and complicated environment is met should as be considered.

**Paper Formatting Concerns:**

No.

**Quality:**

3

**Strengths And Weaknesses:**

Strength:
  The paper further promotes the safety requirements and promotes corresponding solution, which demonstrates outstanding performance.
  The proposed method owns good scalability and is able to deal with conditions with large amounts of agents.

Weakness:
  The paper is a little difficult to understand, for example, it refers to some equations that appear later or even in appendix, which makes the reading not smooth.
  Similar to above, maybe the content is too substantial so that many processes that help understand the work is put in appendix, and the analysis of experiments is also too brief in the main body of the paper. Even the lidar environment is not mentioned in the main body. I think the structure of the article can be improved.
  There are also some typos, such as line 140.

---

> ### Author Rebuttal · Authors · 2025-07-31
>
> # **Response to reviewer MQj4:**
> We thank the reviewer for their positive comments, specifically commending that in complex environments with a large number of agents, our method manages to achieve good performance. We address the issues pointed out by the reviewer below:
>
> ---
>
> **Questions**
>
> ---
>
> > **1**. As is mentioned in previous works (such as HATRPO and MAFOCOPS), the agents will affect each other. Although the proposed method in this paper can deal with multiple agents, it seems to mainly consider each single agent. I’m curious whether this will lead to some instability during training.
>
> Our proposed hierarchical approach involves learning a joint coordination policy between the agents for the high level and the individual skills of the agents at the low level. Given that we are dealing with multi agent policy learning at the high level, we adopt the Centralized Training with Decentralized Execution (CTDE) framework for our high level policy learning. In this approach, agents utilize additional global state information during training to enable centralized learning and improve stability. At execution time, however, each agent’s policy operates solely based on its own local observations, ensuring fully decentralized decision-making. Therefore this schemes ensure the stability during the training. It is worth mentioning that, despite decentralized execution of our high-level policy, our objective is to learn a joint policy for all agents to optimize the overall task objective at the high level. Subsequently, the low level policy is learned conditioned on the high-level action.
>
> > **2**. As the method relies on the CBF and hierarchical skill learning, which I think both require some prior knowledge.
>
> For enforcing our CBF constraints, we require only an approximate dynamics model. That said, emerging differentiable‑simulation frameworks (e.g., JAX + Diffrax, Brax, MJX) can learn such surrogates online, eliminating the manual dynamics‑derivation burden in the future. For hierarchical skill learning, we require only high-level task-related knowledge to define the skill set (specifically, the termination set of a skill). Such prior knowledge is generally available. For example, in our driving environment, the chosen skills are defined intuitively.
>
>
> > **3**. From the paper, I think the authors are very familiar with the experimental environments. But if a new or very complex environment is encountered, how will the method perform?
>
> It is noteworthy that our choice of the MetaDrive simulation environment was motivated by two factors: first, it is a widely used benchmark for multi-agent reinforcement learning, enabling fair comparisons with prior work; second, it allows us to evaluate the effectiveness of our approach on multi-agent safety-critical tasks. In particular, MetaDrive offers such features in two ways: (i) pointwise safety constraints are directly relevant to driving tasks, as there are safety constraints that must be satisfied at all times (e.g., lane-boundary constraints), and (ii) simulation of a large number of agents to assess the scalability of our proposed method. We also implemented our algorithm in a LiDAR-based environment different from the driving task (Appendix A.2), and experimental results validate the performance of our approach. Finally,  if a new and very complex environment is encountered, we anticipate a small upfront effort of defining the skill set.
>
>
>
> > **4**. In the METADRIVE experiments, the chosen baselines do not seem to be algorithms related to safety constraints. How will other methods concerned with safety perform in the environment?
>
> Our work is the first safe hierarchical MARL method (vs. safe MARL, hierarchical MARL, and safe hierarchical control for multi-agent systems). Hence, we implemented the SOTA algorithms that have been applied to MetaDrive, except TRACO [1] (whose code is not publicly available), to validate our approach. We further implemented two SOTA safe MARL algorithms, namely MAPPO-Lagrangian and Multi-Agent Constrained Policy Optimization based on [2], to compare against our approach. Their results are summarized in the table below.
>
> The experiments were run for the bottleneck environment for 2 seeds and the results are summarized in the table below. The results highlight the sample efficiency of our proposed method as well as our safe skills based HRL approach achieves the highest success rate compared to the baselines).
> |Method|Success Rate|Avg. Travel Time (Success rate weighted)|Training Steps|
> |------|-------------:|-----------------:|-------------:|
> |**HMARL-CBF**|$0.99\pm0.00$|$241.16\pm7.37$|$300$K|
> |**MAPPO Lagr** (safety cost $-0.5$)  |  $0.1 \pm 0.05$  | $-$     |  $10$M   |
> |**MAPPO Lagr** (cost threshold: $-2$)  |  $0.22 \pm 0.02$  | $1625.47 \pm 400$     |  $10$M   |
> |**MACPO** (cost threshold:  $-0.5$) |   $0.15 \pm 0.03$  |  $-$   | $10$M   |
> |**MACPO** (cost threshold: $-2$) |   $0.29 \pm 0.02$  |  $621.51 \pm 91.31$   | $10$M   |
>
> **Weaknesses**
>
> ---
>
> > **1**. The paper is a little difficult to understand; for example, it refers to some equations that appear later or even in the appendix, which makes the reading not smooth. Similar to above, maybe the content is too substantial, so many processes that help understand the work are put in the appendix.
>
> We will revise the paper adding details of the supplementary materials (mainly derivations and proofs) and explaining how they support the main results presented in the paper. Additionally, we will bring the definition of the state value, state-action value and advantage value functions given in equations (25) - (27) to explain how the policy gradient for the low level skills can be implemented to learn the safe skills policies.
>
> ---
>
> > **2**. The analysis of experiments is also too brief in the main body of the paper. Even the LiDAR environment is not mentioned in the main body. I think the structure of the article can be improved. There are also some typos, such as line 140.
>
> Unfortunately, due to the strict page limit, we had to move some materials, including the LiDAR environment, to the appendix. However, we are happy to move it back to the main body of the paper if requested. In the final submission, we will also elaborate on the experimental results, including both the evaluation and ablation studies. In particular, we will highlight the distinctions between the environments and clarify the LiDAR-based results.
>
>
> ## Limitations
>
> > **1**. The authors claim to discuss the limitations in both the conclusion and appendix sections, but I don't find the discussion.
>
> Apologies for this. We will expand the discussion of limitations in the conclusion of the final submission, as summarized here. At present, we assume the existence of a predefined skill set for the agents. In the future, we will investigate methods to reduce reliance on task-specific prior knowledge. To this end, we plan to explore skill discovery, allowing agents to identify task-relevant skills autonomously. Alternatively, we will examine skill pruning, where we begin with a large set of general skills and prune it down to the minimal subset required for the task.
>
> > **2** I think the authors should analyze the computational complexity, as the proposed method is based on a hierarchical framework, which tends to be more complex. Also, how the method performs if a new and complicated environment is encountered should be considered.
>
> Because we employ a CTDE scheme, the policies are executed fully decentralized in the real world. Each agent evaluates only two neural networks per control step—the high-level skill selector and the low-level skill policy—which is computationally lightweight on modern GPU-enabled devices. For example, on our workstation equipped with an RTX 4090, we can evaluate the learned policy on up to 45 vehicles in real time, as shown in the submitted videos included with the supplementary material.
>
> ## References ##
> [1] Liu, W., Jing, W., Gao, L., Guo, K., Xu, G., & Liu, Y. (2023). TraCo: Learning Virtual Traffic Coordinator for Cooperation with Multi-Agent Reinforcement Learning. 7th Annual Conference on Robot Learning (CoRL 2023).
>
> [2] Shangding Gu, Jakub Grudzien Kuba, Munning Wen, R. Chen, Z. Wang, Z. Tian, J. Wang, A. Knoll, Y. Yang, "Multi-Agent Constrained Policy Optimisation," arXiv preprint arXiv:2110.02793, 2022.

---

> ### Author Response · Authors · 2025-08-04
> **Re-evaluation and further discussion**
>
> We will like to thank you once again for the thoughtful comments and feedback. We hope our response has addressed your questions and concerns. We will greatly appreciate if you could share any further questions and comments you have and accordingly re-evaluate our submission based on the response.
>
> Thanking you, \
> Authors

---

> > ### Comment · Reviewer_MQj4 · 2025-08-05
> > **Response**
> >
> > Thanks for the response to my review and it solves the majority of my questions. I still recommend the authors to carefully revise the writing and structure of the paper to make it more understandable. But considering the outstanding performance of the algorithm in environments with large amount of agents, I would like to raise my score.

---

> ### Author Response · Authors · 2025-08-05
> **Response**
>
> Thank you very much for your insightful comments and feedback. We are pleased that our revisions addressed your concerns  and thank you for raising the score. We will incorporate your comments in the final submission. Specifically, we will revise the structure and add further explanations of the propositions to clarify how they relate to the main results.
>
> Thank you, \
> Authors

---

### Official Review · Reviewer_Ysva · 2025-07-03

**Clarity:** 2
**Significance:** 3
**Originality:** 2
**Rating:** 3
**Confidence:** 4

**Summary:**

This paper proposes HMARL-CBF, a hierarchical framework for cooperative safety-MARL. The approach decomposes the policy into a high-level policy and a low-level policy. The high-level policy is a policy that selects the skill to use, and the low-level policy is a policy that selects the low-level action with safety constraints conditioned on the skill. The high-level policy is trained by standard MARL methods like MAPPO or QMIX, with the environment reward with a safety penalty. The low-level policy is trained by a CBF-based method with a hand-crafted intrinsic reward. The authors evaluate the proposed method on MetaDrive, and the results show that the proposed method can achieve significant better performance than the baseline methods.

**Questions:**

- It seems that some baselines are not included in Related Work and Experiments sections. Can you provide more comparison with the different methods, including [1,2,3]?
- Can you provide more experiments on different tasks, including [4] and https://safety-gymnasium.readthedocs.io/en/latest/environments/safe_multi_agent.html.
- HMARL-CBF employs a lot of task-specific prior knowledge or ad hoc design, such as the known dynamics of the environment, the design of intrinsic rewards and predefined skill sets. Could you please list in detail the task-related prior knowledge used in this paper? Would this make the comparison with the baseline unfair?
- Could you visualize or at least describe the low-level skills, like what does the skills look like, how long does the skill last, etc.? This would be very helpful for understanding the method presented in this paper.

[1] Berducci L, Yang S, Mangharam R, et al. Learning adaptive safety for multi-agent systems[C]//2024 IEEE International Conference on Robotics and Automation (ICRA). IEEE, 2024: 2859-2865.

[2] Zhang S, So O, Black M, et al. Discrete GCBF Proximal Policy Optimization for Multi-agent Safe Optimal Control[J]. arXiv preprint arXiv:2502.03640, 2025.

[3] Gu S, Kuba J G, Wen M, et al. Multi-agent constrained policy optimisation[J]. arXiv preprint arXiv:2110.02793, 2021.

[4] O’Kelly, M., Zheng, H., Karthik, D. &amp; Mangharam, R.. (2020). F1TENTH: An Open-source Evaluation Environment for Continuous Control and Reinforcement Learning. <i>Proceedings of the NeurIPS 2019 Competition and Demonstration Track</i>, in <i>Proceedings of Machine Learning Research</i> 123:77-89 Available from https://proceedings.mlr.press/v123/o-kelly20a.html.

**Ethical Concerns:**

["NO or VERY MINOR ethics concerns only"]

**Final Justification:**

The authors’ response has partially addressed my concerns. However, I still believe that:

1. The generality of the task-specific designs in this paper (e.g., predefined skills, intrinsic rewards, Lyapunov functions) remains questionable. These ad hoc components seem difficult to generalize to new environments. The authors should devote more space in the main text to clearly describe the assumptions and task-specific designs they adopt, and discuss their applicability and scalability.

2. As some other reviewers have also noted, the presentation of the paper could be improved.

Therefore, I have decided to keep my rating 3 (borderline rejection).

The method shows promising results on the chosen benchmarks, which suggests its potential. However, I encourage the authors to conduct more in-depth studies and experiments to gain a deeper understanding of why the method works and the scope of its applicability.

**Limitations:**

Yes

**Quality:**

2

**Strengths And Weaknesses:**

### Strengths
- The performance of the proposed method is impressive on MetaDrive.
- Hierarchical policy is a good approach to handle the large and complex environment of MARL.

### Weakness
- The writing is not clear and has some typos, e.g.
    - line 74: [46] is not a multi-agent RL method.
    - line 138: $s^i, s^j$ seem to be redundant.
    - line 140: "hus" should be "Thus".
    - line 165: "An skill" should be "A skill".
    - eq (8), line 182, eq (10)...: The condition should be put outside the reward function and inside the expectation, or the definition of the reward function should be changed.
    - eq (9): $p_j^i$ is not defined.
    - eq (9): "for some $i$" should be removed.
    - footnote 1 on page 5: $\mathbb {I}(x)$ is a binary function by definition, while in this footnote it is claimed to be a large number $M$, which is unclear.
    - eq (14): Notations like $\Omega, \zeta$ are not defined.
- The novelty to use CBF-based method to train the low-level policy in MARL is limited.
- The approach seem to require a lot of artificial design, e.g. the hand-crafted intrinsic reward, the predefined skill set, the ad hoc Lyapunov function, etc.
- The experimental results are impressive, but the chosen tasks are relatively few. Also, some baseline methods are not included in the comparison.

---

> ### Author Rebuttal · Authors · 2025-07-31
>
> # **Response to reviewer Ysva:**
> We thank the reviewer for their positive comments and highlighting that "hierarchical policy is a good approach to handle the large and complex environment of MARL" and the results of our "proposed method is impressive". We are thankful for the constructive criticism. We address the issues pointed out by the reviewer below.
>
> ## Summary
>
> We provide the summary of the comments addressed in the response below:
>
> - Provided further baseline comparisons against SOTA Safe MARL algorithms.
> - Provided clarification on our proposed Safe Hierarchical MARL framework.
> - Provided additional clarification on the implementation of our proposed algorithm.
>
> ## Questions
>
> > **Q1**. It seems that some baselines are not included in Related Work and Experiments sections. Can you provide more comparison with the different methods, including [1,2,3]?
>
> Thank you for the feedback. We will incorporate additional literature in the final submission. Based on our survey, our paper is the first work on safe multi-agent HRL. Hence, we compared our method against the state of the art baselines in MetaDrive environment. Our experiments on MetaDrive offers manifold benefits, namely: (i) simulate large number of physical agents, (ii) incorporate pointwise in time safety‑critical interactions between agents. We ran MACPO [3] (Multi agent Constrained Policy Optimization) and MAPPO Langrangian [3] algorithm for the bottleneck environment in MetaDrive. The results are summarized in the table below.
> |Method|Success Rate|Avg. Travel Time (Success rate weighted)|Training Steps|
> |------|-------------:|-----------------:|-------------:|
> |**HMARL-CBF**|$0.99\pm0.00$|$241.16\pm7.37$|$300$K|
> |**MAPPO Lagr** (safety cost $-0.5$)  |  $0.1 \pm 0.05$  | $-$     |  $10$M   |
> |**MAPPO Lagr** (cost threshold: $-2$)  |  $0.22 \pm 0.02$  | $1625.47 \pm 400$     |  $10$M   |
> |**MACPO** (cost threshold:  $-0.5$) |   $0.15 \pm 0.03$  |  $-$   | $10$M   |
> |**MACPO** (cost threshold: $-2$) |   $0.29 \pm 0.02$  |  $621.51 \pm 91.31$   | $10$M   |
>
> Unfortunately, MetaDrive environment is not built using JAX which is the library used for implementing the DGPPO (Discrete Graph CBF PPO) [2] algorithm. Therefore, the experiment runs very slowly. Each rollout of 1000 steps takes approximately 20 minutes on a server equipped with RTX 4050 GPU. Therefore, it would around a week to train the algorithm for 1M steps. Thus, we run the algorithm on a suite of lidar environments provided by the authors of DGPPO [2] along with MAPPO Lagrangian algorithm [3]. The number of agents The results are summarized in the table below.
>
> |Method|LidarTarget|LidarSpread|LidarLine|Training Steps|
> |------|-------------:|-----------------:|-------------:|--------------:|
> |**DGPPO**|  $0.98\pm 0.00$ | $0.99 \pm 0.00$ | $0.99 \pm 0.00$ |1M|
> |**MAPPO Lagr_5**| $0.99 \pm 0.00$ | $0.99 \pm 0.00$ | $1 \pm 0.00$|1M|
> |**HMARL-CBF**| $0.99 \pm 0.00$ | $0.99 \pm 0.00$ | $.98 \pm 0.02$|300k|
>
> The presented results highlight the sample efficiency provided by our hierarchical approach. Besides that, our approach achieves outperforms the baselines in MetaDrive bottleneck environment where the number of agents was set to 25, and similar results in the LiDAR environment suite where the agent count was set to the default value of 3. Finally, [1] proposes a method for safe single-agent RL based algorithms for solving stochastic games and validates it in multi-agent settings.
>
> ---
>
> > **Q2**. Can you provide more experiments on different tasks, including [4] and Safety-gymnasium.
>
> Please refer to **Q1** for our response. We have implemented additional baselines and also ran our algorithm on environments from the recent SOTA paper [2] to validate our proposed approach.
>
> ---
>
> > **Q3**. Knowledge of dynamics.
>
> Great question! In fact, we require only an **approximate** model of the system for computing the time derivative that appears in the CBF constraints. In our experiments, we employ the standard kinematic bicycle model as an approximation for the MetaDrive environment, which is built on the PyBullet physics engine which simulates the vehicle dynamics. The presented results in table 1 validates our approach.
>
> ---
>
> > **Q4** Intrinsic rewards and predefined skill sets.
>
> The skills are defined from high level task related prior knowledge. However, our approach poses no limitation on the cardinality of the skill set. In our proposed HRL approach,  the low level skills are not pretrained. We define the skills as stochastic constrained optimal control problem. The intrinsic reward is defined as the norm of the difference between the initiation state and skill termination state; thus can be generically extended to any skill. And, the high level coordination policy and the low level skills are both learned during training jointly. This allows us to define skills that are interpretable and transferable to other similar environments as evident from the generalization results in Appendix section A.2.3, Table 5.
>
> ---
>
> > **Q4**. Could you visualize or at least describe the low-level skills, like what does the skills look like, how long does the skill last, etc.? This would be very helpful for understanding the method presented in this paper.
>
> We define the skills as fixed final state stochastic OCP which have semantic interpretation. Hence, the length of the skills is a variable. For example, in MetaDrive environment we define the skills to be left and right lane changing, cruising, speeding up and slowing down. Each of these skills take different length of time to execute. Besides that the skills policy and therefore the length of the skill is also dependent on the behavior of the other agents observable to an agent.
>
> ## Weakness
> > 1. Typos and notations.
>
> Thank you for pointing this out. We will revise the typos and notational issues in the final submission.
>
> ---
>
> > 2. The novelty to use CBF-based method to train the low-level policy in MARL is limited.
>
> We would like to add that based on our survey, our paper is the first work on Safe Hierarchical MARL for multi-agent safety-critical systems. Our posed hierarchical CBF based safe skill approach achieves high success rate as evident from the evaluation and baseline comparison results in safety-critical systems.
>
> ---
>
> > 3. The approach seem to require a lot of artificial design, e.g. the hand-crafted intrinsic reward, the predefined skill set, the ad hoc Lyapunov function, etc.
> Add explanation
>
> Please refer to **Q3** and **Q4** for the response.
>
> ---
>
> > 4. The experimental results are impressive, but the chosen tasks are relatively few. Also, some baseline methods are not included in the comparison.
>
> Please refer to **Q1** for the response.
>
> ## References ##
> [1] Berducci L, Yang S, Mangharam R, et al. Learning adaptive safety for multi-agent systems[C]//2024 IEEE International Conference on Robotics and Automation (ICRA). IEEE, 2024: 2859-2865.
>
> [2] Zhang S, So O, Black M, et al. Discrete GCBF Proximal Policy Optimization for Multi-agent Safe Optimal Control[J]. arXiv preprint arXiv:2502.03640, 2025.
>
> [3] Shangding Gu, Jakub Grudzien Kuba, Munning Wen, R. Chen, Z. Wang, Z. Tian, J. Wang, A. Knoll, Y. Yang, "Multi-Agent Constrained Policy Optimisation," arXiv preprint arXiv:2110.02793, 2022.

---

> ### Author Response · Authors · 2025-08-04
> **Re-evaluation and further discussion**
>
> We will like to thank you once again for the thoughtful comments and feedback. We hope our response has addressed your questions and concerns. We will greatly appreciate if you could share any further questions and comments you have and accordingly re-evaluate our submission based on the response.
>
> Thanking you, \
> Authors

---

> > ### Comment · Reviewer_Ysva · 2025-08-06
> >
> > The authors’ response has partially addressed my concerns. However, I still believe that:
> >
> > 1. The generality of the task-specific designs in this paper (e.g., predefined skills, intrinsic rewards, Lyapunov functions) remains questionable. These ad hoc components seem difficult to generalize to new environments. The authors should devote more space in the main text to clearly describe the assumptions and task-specific designs they adopt, and discuss their applicability and scalability.
> >
> > 2. As some other reviewers have also noted, the presentation of the paper could be improved.
> >
> > Therefore, I have decided to keep my rating unchanged.
> >
> > The method shows promising results on the chosen benchmarks, which suggests its potential. However, I encourage the authors to conduct more in-depth studies and experiments to gain a deeper understanding of why the method works and the scope of its applicability.

---

> ### Author Response · Authors · 2025-08-06
> **Response to Official Comment by Reviewer Ysva**
>
> We will like to thank the reviewer for their response and acknowledging the merit of our method based on the presented results and additional baselines. Please find our response summarized below.
>
> ## Questions
>
> ---
>
> > **Q1** The generality of the task-specific designs in this paper (e.g., predefined skills, intrinsic rewards, Lyapunov functions) remains questionable. These ad hoc components seem difficult to generalize to new environments. The authors should devote more space in the main text to clearly describe the assumptions and task-specific designs they adopt, and discuss their applicability and scalability.
>
> ---
>
> Unfortunately, we were cramped for room to further elaborate on these remarks in the main paper. However, we will revise the final draft by adding further explanation on: i. the skill-related assumptions and ii. task-specific skills (which are currently described in Appendix section A.5) in the main body of the paper.
>
> Although our skills are predefined, we do not place any limitation or restriction on the cardinality of the skill set. Additionally, the only component that is predefined in a skill is its termination condition. All the other components, including the intrinsic reward and the control Lyapunov function, are automatically derived or learned.
>
> ---
>
> > **Q2** As some other reviewers have also noted, the presentation of the paper could be improved.
>
> ---
>
> We will revise the paper to improve the clarity based on all the reviewers' comments. Please let us know any specific comments you have in this regard, and we will accordingly incorporate them in the revision.
>
> ---
>
> Please let us know your further comments and questions, and we will address them. Based on the previously provided baseline comparison results and the clarifications, we will greatly appreciate if you could consider revising your score.

---

### Official Review · Reviewer_1MFt · 2025-07-03

**Clarity:** 1
**Significance:** 2
**Originality:** 3
**Rating:** 3
**Confidence:** 2

**Summary:**

This paper introduces a hierarchical multi-agent reinforcement learning framework for learning safe policies based on control barrier functions. The control problem is formulated as a partially-observable multi-agent MDP with pointwise-in-time hard constraints based on barrier functions. The proposed framework learns high-level control policies that select skills, considering performance, while the low-level control policies implement skills by selecting actions, considering strict safety constraints (policy learning is based on existing multi- and single-agent algorithms, such as MAPPO and PPO). The proposed framework is evaluated in the MetaDrive simulator with various scenarios, showing that compared to multi-agent baselines the proposed framework provides better satisfaction of safety requirements.

**Questions:**

* Where do skills come from?
* Are the baseline methods SOTA in safe RL?

**Ethical Concerns:**

["NO or VERY MINOR ethics concerns only"]

**Final Justification:**

The paper makes an interesting contribution to safe multi-agent reinforcement learning, and the proposed framework appears to be promising in terms of numerical results. However, the presentation of the framework is not very clear (as other reviewers also noted), and the paper would benefit from a major revision, which should also include the additional results presented in the rebuttals.

**Limitations:**

Yes.

**Paper Formatting Concerns:**

Font size in Figure 2 is very small.

**Quality:**

2

**Strengths And Weaknesses:**

Strengths:
* Safety-critical multi-agent control is an important problem.
* The proposed hierarchical decomposition into high-level and safe low-level policies is promising.

Weaknesses:
* Some aspects of the proposed framework are not clear. Where do the skills come from? The high-level policy seems to learn to select from a given set of skills, while the low-level policies seem to be learned for specific skill. Where do these skills come from in the first place?
* The baseline approaches in the numerical experiments do not seem to be designed for safety (they seem to be generic multi-agent RL approaches). Do these really represent the SOTA in safe multi-agent RL?
* "While CMDPs ensure safety over trajectories, they do not consider pointwise time constraints critical for safety-critical systems."
Is this really a limitation? One could define (1) a cost function that is non-zero for pointwise time-constraint violation and (2) cost constraint that enforces zero expected cost. Would that not capture pointwise constraints?
* The description of termination strategies on lines 94 to 98 is rather informal, and it is not clear how the $\tau_{continue}$ strategy is implemented in the model.
* The notation on line 133 seems to be inconsistent: barrier function $b$ first takes agent-specific states as input, but then the specification of its domain includes the entire state space $\mathcal{S}$.
* The definition of the observation space on lines 137 and 138 is strange: the observation space is defined as a superset of the state space, but the environment is partially observable, which should be modeled as some states being indistinguishable to the agent, implying that the observation space is a subset of the state space. But this of course would prevent applying the barrier function $b$ to the observation. Can the barrier function really be evaluated on observations instead of states? This seems to contradict partial observability if the barrier function are general functions of state.
* On lines 149 to 152, the discussion emphasizes relaxing the assumption of CMDP that policies depend on the joint state. But this does not seem significant, it is just a simple matter of definition.
* The definition of the indicator function at the bottom of page 5 has some issues. The clause "for some $i$" does not make sense since $i$ is given as the running variable of the first summation of Equation (9), hence it makes not sense to refer to "some $i$" (also, within the definition of the indicator function the domain of $i$ would be unclear). Footnote 1 is unclear since it suggests that the indicator function would output some large value $M$, but according to the definition, it outputs 0 or 1.
* Calling the proposed framework "Safety Guaranteed" might be misleading since safety is not guaranteed, it is achieved only high probability.
* It may help readability if the distribution of random variable in the subscript of expectations is specified, e.g., in Equation (1), clarify that $z \sim \pi$ and $k, s' \sim P$.

---

> ### Author Rebuttal · Authors · 2025-07-31
>
> # **Response to reviewer 1MFt:**
> We thank the reviewer for their positive comments that the addressed "safety-critical multi agent control is an important problem" and that our proposed "hierarchical approach is promising".  We also extend our gratitude for the constructive criticism. We address the issues pointed out by the reviewer below.
>
> ## Summary ##
>
> The summary of the points addressed in the reponse is provided below:
>
> - Provide explaination of the .
> - Provide additional baseline comparisons against existing safe MARL approaches
> - Provide clarifications for the reviewers comments.
>
> ## Details: ##
>
> **Questions:**
> > 1. Where do skills come from?
>
> The skills are defined from high level task related prior knowledge. However, our approach poses no limitation on the cardinality of the skill set. In our proposed HRL approach,  the low level skills are not pretrained. We define the skills as stochastic constrained optimal control problem. And, the high level coordination and the low level skills both are learned during training jointly. This allows us to define skills that are interpretable and transferable to other similar environments as evident from the generalization results in Appendix section A.2.3, Table 5. Based on the definition of the skill set, our method learns the policy over the skill sets to accomplish the task cooperatively and safely in multi-agent environment.
>
> > 2. Are the baseline methods SOTA in safe RL?
>
> Based on our survey, our paper is the first work on safe multi-agent HRL. Hence, we compared our method against the state of the art baselines in MetaDrive environment. The provided results highlight the merit of our proposed approach. Regarding safe MARL baseline: we ran MACPO [1] (Multi agent Constrained Policy Optimization) and MAPPO Langrangian [2] algorithms. The experiments were run for the bottleneck environment for 2 seeds and the results are summarized in the table below. The results highlight the sample efficiency of our proposed method as well as our safe skills based HRL approach achieves the highest success rate compared to the baselines).
> |Method|Success Rate|Avg. Travel Time (Success rate weighted)|Training Steps|
> |------|-------------:|-----------------:|-------------:|
> |**HMARL-CBF**|$0.99\pm0.00$|$241.16\pm7.37$|$300$K|
> |**MAPPO Lagr** (safety cost $-0.5$)  |  $0.1 \pm 0.05$  | $-$     |  $10$M   |
> |**MAPPO Lagr** (cost threshold: $-2$)  |  $0.22 \pm 0.02$  | $1625.47 \pm 400$     |  $10$M   |
> |**MACPO** (cost threshold:  $-0.5$) |   $0.15 \pm 0.03$  |  $-$   | $10$M   |
> |**MACPO** (cost threshold: $-2$) |   $0.29 \pm 0.02$  |  $621.51 \pm 91.31$   | $10$M   |
>
> **Weaknesses:**
> > 1. Some aspects of the proposed framework are not clear. Where do the skills come from? The high-level policy seems to learn to select from a given set of skills, while the low-level policies seem to be learned for specific skill. Where do these skills come from in the first place
>
> We use only high-level task-related prior knowledge to define the skill set. Such prior knowledge is generally available. For example, in our MetaDrive driving environment the chosen skills are defined intuitively which include left and right lane changing, cruising, throttling and slowing down. We do not enforce any limit on the number of skills, nor, the diversity of the skills set. Finally, we jointly learn both the high- and low-level policies.
>
>
> > 2. The baseline approaches in the numerical experiments do not seem to be designed for safety (they seem to be generic multi-agent RL approaches). Do these really represent the SOTA in safe multi-agent RL?
>
> Please refer to **Q2** for the response.
>
>
> > 3. "While CMDPs ensure safety over trajectories, they do not consider pointwise time constraints critical for safety-critical systems." Is this really a limitation? One could define (1) a cost function that is non-zero for pointwise time-constraint violation and (2) cost constraint that enforces zero expected cost. Would that not capture pointwise constraints?
>
> Thanks for making this great point. CMDPs consider discounted sum of costs in the infinite horizon problem scenario and applies a cost threshold on the expected sum of discounted cost. Firstly, taking the discounted sum poses less emphasis on long time cost violation. Besides that, as CMDP enforces a safe cost threshold on the expectation of trajectories, therefore, it places less emphasis on cost violation of trajectories with lower probability. Finally, this way of cost definition is susceptible to training instability particularly for primal dual methods.
>
> > 4. The description of termination strategies on lines 94 to 98 is rather informal, and it is not clear how the $\tau_{continuous}$ strategy is implemented in the model.
>
> The $\tau_{continue}$ is an asynchronous skill selection scheme. Under this scheme, each agent selects and executes the skill until completion. Following that the agent picks a new skill and this process is repeated by every agent in the multi-agent setting. Thus, the decision epoch is defined as the time when any of the joint skills of the agents terminate in our setting.
>
> > 5. The notation on line 133 seems to be inconsistent: barrier function $b$ first takes agent-specific states as input, but then the specification of its domain includes the entire state space $\mathcal{S}$. The definition of the observation space on lines 137 and 138 is strange: the observation space is defined as a superset of the state space, but the environment is partially observable, which should be modeled as some states being indistinguishable to the agent, implying that the observation space is a subset of the state space. But this of course would prevent applying the barrier function $b$ to the observation. Can the barrier function really be evaluated on observations instead of states? This seems to contradict partial observability if the barrier function are general functions of state.
>
> Thank you for pointing this out. We have revised it in the paper. The joint state space is defined as $\mathcal{S} = \mathcal{S}^{e} \times \prod_{i\in\mathcal{N}} \mathcal{S}^{i}$ where $\mathcal{S}^{e}$ denotes the environment state space and $\mathcal{S}^{i}$ the state space of agent $i$. Each agent $i$ receives an observation $o^{i} \in \mathcal{O}^{i} \subseteq \mathcal{S}$, which includes its own state $s^{i}$ together with those portions of other agents’ or environmental states that are observable to it. Safety is encoded by a control‑barrier function $b : \mathcal{O}^{i} \to \mathbb{R}$; equivalently, one may write $b(s^{i}, s^{j})$ to emphasize the dependence on agent $i$’s state and the observed state $s^{j}$ of another entity. The safety constraint is expressed in the form $b(o^{i}) \ge 0$.
>
>
> > 7. On lines 149 to 152, the discussion emphasizes relaxing the assumption of CMDP that policies depend on the joint state. But this does not seem significant, it is just a simple matter of definition.
>
> If the constraints depend on the joint state then during execution the agents have to receive information of all other agents that are not observable to it. Executing such a policy in a real multi-agent system demands a central hub (or all-to-all communication) at each control step.
>
> > 8. The definition of the indicator function at the bottom of page 5 has some issues. The clause "for some $i$" does not make sense since $i$ is given as the running variable of the first summation of Equation (9), hence it makes not sense to refer to "some $i$" (also, within the definition of the indicator function the domain of $i$ would be unclear). Footnote 1 is unclear since it suggests that the indicator function would output some large value $M$, but according to the definition, it outputs 0 or 1.
>
> Thank you for pointing this out. We will remove this footnote from the final submission.
>
> > 9. Calling the proposed framework "Safety Guaranteed" might be misleading since safety is not guaranteed, it is achieved only high probability.
>
> We will revise the introduction and explicitly mention the conditions under which our proposed framework can provide safety guarantee through safe execution of the low level skills.
>
> > 10. It may help readability if the distribution of random variable in the subscript of expectations is specified, e.g., in Equation (1), clarify that $z\sim \pi$ and $k, s' \sim P$.
>
> Thank you for sharing this feedback. We will incorporate this comment in the final submission.
>
> ---
>
> ## References
>
> [1] Shangding Gu, Jakub Grudzien Kuba, Munning Wen, R. Chen, Z. Wang, Z. Tian, J. Wang, A. Knoll, Y. Yang, "Multi-Agent Constrained Policy Optimisation," arXiv preprint arXiv:2110.02793, 2022.

---

> ### Author Response · Authors · 2025-08-04
> **Re-evaluation and further discussion**
>
> We will like to thank you once again for the thoughtful comments and feedback. We hope our response has addressed your questions and concerns. We will greatly appreciate if you could share any further questions and comments you have and accordingly re-evaluate our submission based on the response.
>
> Thanking you, \
> Authors

---

> > ### Comment · Reviewer_1MFt · 2025-08-06
> >
> > Thank you for the detailed response. It has answered most of my questions. It will be great if the authors can incorporate these clarifications into the paper, such as clarifying that skills are exogenously given.
> >
> > Regarding the baselines, I still have some concerns. The authors write that "Based on our survey, our paper is the first work on safe multi-agent HRL." While I agree that the paper does appear to be the first one to consider safe **hierarchical** multi-agent RL, it is not the first safe multi-agent RL framework. Are non-hierarchical multi-agent RL approaches inapplicable as baselines? If the emphasis is on hierarchical multi-agent RL, why are the baselines not hierarchical?
> >
> > The paper presents promising results, but as other reviewers have also noted, the presentation is not particularly clear. I am afraid that the text would need a rather significant revision.

---

> ### Author Response · Authors · 2025-08-06
> **Response to Official Comment by Reviewer 1MFt**
>
> Thank you very much for your response. We provide below the response to your further queries.
>
> ## Questions
>
> ---
>
> > **Q1** It will be great if the authors can incorporate these clarifications into the paper, such as clarifying that skills are exogenously given.
>
> ---
>
> We will incorporate the clarifications including: i. adding skills are exogenously given, ii. the description of skill termination strategy, iii. the notations associated to the barrier function $b$, iv. removal of the indicator function on page 5, v. revising the notations in the equations including equation (1), in the final submission. Additionally, we will revise the remainder of the paper to improve clarity based on all the reviewers' feedback.
>
> ---
>
> > **Q2** Regarding the baselines, I still have some concerns. The authors write that "Based on our survey, our paper is the first work on safe multi-agent HRL." While I agree that the paper does appear to be the first one to consider safe hierarchical multi-agent RL, it is not the first safe multi-agent RL framework. Are non-hierarchical multi-agent RL approaches inapplicable as baselines? If the emphasis is on hierarchical multi-agent RL, why are the baselines not hierarchical?
>
> ---
>
> We would like to clarify that, as our work is the first one to consider safe hierarchical multi-agent RL, we have therefore indeed considered flat policy learning baselines, including the state-of-the-art (SOTA) baselines available for the particular environment and safe multi-agent RL baselines. We have validated our algorithm against the baselines in two environment suites, namely i. MetaDrive and ii. LiDAR environment suites. Please find the summary of the baselines below:
>
> | Algo. Type | Algo. Name |
> |------------|------------|
> | RL| IPPO |
> |MARL| MFPO, CL, CoPo|
> | Safe MARL  | MAPPO Lagrangian [1], MACPO [1], DGPPO [2] |
>
> The results presented in the response, as well as in the paper, highlight the sample efficiency achieved through our hierarchical policy decomposition method. Furthermore, our safe skills are able to achieve near-perfect success rates across the two suites of environments.
>
> Please share your further questions and concerns, and we will address them. We will be thankful if you can revise your score in the light of our given response.
>
> ---
>
> ## References
> [1] Shangding Gu, Jakub Grudzien Kuba, Munning Wen, R. Chen, Z. Wang, Z. Tian, J. Wang, A. Knoll, Y. Yang, "Multi-Agent Constrained Policy Optimisation," arXiv preprint arXiv:2110.02793, 2022
>
> [2] Zhang S, So O, Black M, Fan C. Discrete GCBF Proximal Policy Optimization for Multi-agent Safe Optimal Control, The Thirteenth International Conference on Learning Representations (ICLR). 2025.

---

> > ### Comment · Reviewer_1MFt · 2025-08-06
> >
> > Thanks for the further clarification on the baselines. I think that it will be very important to incorporate the safe MARL results into the main paper.

---

> > > ### Author Response · Authors · 2025-08-06
> > > **Response to Official Comment by Reviewer 1MFt**
> > >
> > > We are thankful for your response and feedback in helping us improving the paper through this discussion.
> > >
> > > Thank you for this great feedback. We will incorporate the results for Safe MARL baselines in the main body of the paper by extending Table 1. Currently, the results are in appendix section. However, we will incorporate all the baseline results shared in the response as well as in the appendix section in the main body of the paper.
> > >
> > > Please let us know your further questions and concerns, and we will gladly address them. We will greatly appreciate if you could consider our discussion to revise your score.
> > >
> > > Thanking you, \
> > > Authors.

---

> > > > ### Author Response · Authors · 2025-08-07
> > > > **Response to Official Comment by Reviewer 1MFt**
> > > >
> > > > We will like to thank you once again for the valuable discussion.
> > > >
> > > > Please let us know if you have any further comments or questions, and we will address them. We will incorporate all the changes you suggested in the final submission. We would greatly appreciate it if you could revise your score in light of our discussion.
> > > >
> > > > Thanking you, \
> > > > Authors

---

> > > > > ### Author Response · Authors · 2025-08-08
> > > > > **Follow up response regarding score revision**
> > > > >
> > > > > Dear Reviewer,
> > > > >
> > > > > Thank you once again for your insightful feedback and helpful discussion. Under the light of the latest response, we will be really thankful if you can **consider raising your score** if we have addressed all your concerns. Please let us know your further thoughts and queries and we will gladly address them.
> > > > >
> > > > > We thank you for your continued time and effort in this regard.
> > > > >
> > > > > Thanks, \
> > > > > Authors

---

### Official Review · Reviewer_17QD · 2025-07-03

**Clarity:** 2
**Significance:** 3
**Originality:** 3
**Rating:** 5
**Confidence:** 4

**Summary:**

The paper introduces HMARL-CBF, a two-level framework for safe multi-agent reinforcement learning in partially-observable settings with hard safety constraints. At the high level a centralized-training decentralized-execution (CTDE) policy selects a joint skill for all agents. These skills are temporally-extended options that encode cooperative behaviour. At the low level each agent executes its chosen skill through a quadratic-program (QP) controller that embeds control-barrier-function (CBF) constraints, guaranteeing point-wise-in-time safety. Experiments in the MetaDrive traffic simulator show near-perfect collision-free success rates (≥ 95 %) and substantially shorter episodes than baselines like IPPO, MFPO, CoPo and a flat MAPPO variant.

**Questions:**

- Did you give baselines access to the same pre-trained skill library (e.g. by augmenting the action space with them) and CBF safety filter? If not, please justify. A MAPPO or IPPO baseline equipped with the identical skill set would isolate the contribution of the hierarchical credit assignment itself. This my main concern from this work.

- Section 5.2 relies on known dynamics for the QP. What happens when only an approximate or learned model is available? Please include experiments with modelling errors if possible.

- Five seeds are unusually low for high-variance MARL benchmarks. Please report 10–20 seeds with 95 % confidence intervals, otherwise this severely impacts the significance of the claims

- Please give intuitive/informal explanations in Section 3.2. This will greatly improve accessibility for the broader NeurIPS audience.

**Ethical Concerns:**

["NO or VERY MINOR ethics concerns only"]

**Final Justification:**

The authors really took some time and effort to properly address my concerns and provide a very detailed response. I am happy with their clarification and happy to increase my score to an accept accordingly.

**Limitations:**

yes

**Quality:**

2

**Strengths And Weaknesses:**

# Strengths

### Quality
The paper presents a technically sophisticated integration of hierarchical reinforcement learning with control-barrier-function (CBF) safety filtering. The methodology is sound: a centralized-training, decentralized-execution scheme selects joint skills, while individual agents execute them through CBF-constrained quadratic-program controllers. Empirical results in the continuous-control MetaDrive suite show dramatic gains in collision-free success rates and episode efficiency, and the authors include ablations that dissect the impact of hierarchy and safety filtering.

### Clarity
The high-level idea and algorithmic pipeline are conveyed effectively through Figure 1 and Section 5, giving readers a coherent narrative from problem statement to implementation. An extensive appendix provides proofs and additional details, which helps knowledgeable readers follow the derivations and design choices.

### Significance
Addressing hard safety in cooperative multi-agent RL is an important open problem with clear relevance to autonomous driving and swarm robotics. Demonstrating near-perfect safety without sacrificing task efficiency could influence future research and practical deployments of MARL systems subject to stringent safety constraints.

### Originality
While action-space augmentation via options and CBF-based filtering are themselves established, the paper’s specific decomposition—first selecting a joint high-level skill and then letting agents independently apply CBF-filtered low-level control—appears novel. This bridges two previously separate literatures (hierarchical MARL and safe control) in a way that could inspire further hybrid approaches.


# Weaknesses

### Quality
The safety guarantees hinge on knowing affine system dynamics (Eq. 3); experiments likewise assume exact vehicle models, leaving robustness to modelling error unexplored. Pre-defined skills are hand-crafted and provided only to the proposed method, so performance gains may stem from privileged prior knowledge. Key baselines such as MAPPO augmented with identical skills and CBF filters are missing. Comparisons are averaged over only five seeds, and no statistical significance tests are reported.

### Clarity
Despite a clear high-level story, numerous notation and referencing issues impede readability. Symbols like $\mathcal{U}$ in Eq. 4 are undefined; Eq. (17)/(18) are cited but absent; time-index notation is inconsistent; acronyms such as QP and LQR are never introduced or done after usage; and Section 3.2 offers little intuition for (HO)CBFs. Typos and un-hyperlinked citations and references further detract from polish. Numbers are also generally floating without being specific on whether they correspond to equations/sections/etc.

Some of the notations in Section 4 are also confusing. E.g. The joint state space $\mathcal{S}$ was defined before in line 87 as just being the joint agent states $\mathcal{S}^i$. Then in line 127 it is defined as the joint agent states including the environment states $\mathcal{S}^e$. Hence it is now unclear what $\mathcal{S}^i \subseteq \mathcal{O}^i$ means (line 137) when the authors have that $\mathcal{O}^i \subseteq \mathcal{S}$. Is $\mathcal{S}^i$ now defined differently? What does $s^i \in \mathcal{O}^i$ mean? Are observations and states now the same? Is $b$ a function of observation, a single agent state, the agent joint states, or the joint states with the environment states (since sometimes it is $b(s)$, $b(o)$, $b(s^i,s^j)$)? In general the precise mathematical formulation in this section is confusing and unclear.

### Significance
The evaluation is confined to MetaDrive traffic scenarios (and toy navigation tasks in the appendix). Generalisation to higher-dimensional platforms (e.g., MuJoCo robots like the Ant) or to settings with learned dynamics is not demonstrated. Until such breadth is shown (or at least argued theoretically) the broader impact on real-world applications remains uncertain.

### Originality
Because both option-based action-space augmentation and CBF safety filtering have established precedents, the contribution’s novelty rests largely on their particular combination and training recipe. The paper could better differentiate itself from prior safe-HRL works to underscore what is truly unique.

---

> ### Author Rebuttal · Authors · 2025-07-31
>
> # **Response to reviewer 17QD:**
>
> We thank the reviewer for acknowledging our method as a sound and technically sophisticated integration of hierarchical RL with CBF safety filtering and our significant experimental results which show dramatic gains in collision-free success rates and episode efficiency. We address the issues pointed out by the reviewer below:
>
> ## Summary
>
> In brief, we address the following:
>
> - Provide additional experiments to compare our method with MAPPO and IPPO with skills and CBF filters.
> - Provide additional experiments to study the influence of modelling errors.
> - Provide additional experimental results with more seeds.
> - Provide more intuition about the use of CBFs.
> - Refine the notations.
> - Highlight our unique contributions.
>
> ## Details
>
> >**Q1**: Did you give baselines access to the same pre-trained skill library (e.g. by augmenting the action space with them) and CBF safety filter? Comparison with MAPPO and IPPO with skills and the CBF safety filter.
>
> In HMARL-CBF each skill is a stochastic, constrained optimal control problem; high-level coordination and low-level skills are learned jointly, not pre-trained. Because MAPPO and IPPO are flat, adding variable-length skills requires inserting a policy hierarchy.  Besides low level skills are learnt to optimize the entire trajectory as opposed to the trajectory corresponding to the skill in order to improve the overall performance. Our skill formulation makes them interpretable, provably safe, and transferable to related environments—an explicit contribution of our work. In Appendix A.2.3, Table 5, we show that skills learned this way are transferrable across similar environments.
>
> Section 6.4 Table 2, presents the ablation “Hierarchical MARL with parameterized LQR‑based skills,” in which skills are incorporated into MAPPO by altering the base algorithm. Each skill is formulated as a parametric LQR problem whose objective steers the agent towards the skill’s termination state. This ablation removes the safety filter from HMARL‑CBF. We also implement MAPPO and IPPO augmented with CBF filters in the bottleneck environment. The MAPPO-LQR approach only adds safety violation penalty to the high and low level rewards which fails as the LQR based skills do not consider the safety constraints. Despite incorporating safety filters both MAPPO and IPPO fails to learn the policies to complete the task which is consistent with our flat policy ablation results in Table 2.
>
> |Method|Success Rate|Avg. Travel Time|Training Steps|
> |------|-------------:|-----------------:|-------------:|
> |**HMARL-CBF**|$0.99\pm0.00$|$214.16\pm7.37$|1M|
> |MAPPO+Parametric LQR skills|$0.00\pm0.00$|--|1M|
> |MAPPO+CBF filter (no skills)|$0.00\pm0.00$|--|1M|
> |IPPO+CBF filter (no skills)|$0.00\pm0.00$|--|1M|
> |MAPPO+Parametric LQR skills+CBF filter|$0.00\pm0.00$|--|1M|
>
> ---
>
> > **Q2**: Section 5.2 relies on known dynamics for the QP. What happens when only an approximate or learned model is available? Please include experiments with modelling errors if possible.
>
> Great question! In fact, we require only an **approximate** model of the system for computing the time derivative that appears in the CBF constraints. In our experiments, we employ the standard kinematic bicycle model as an approximation for the MetaDrive environment, which is built on the PyBullet physics engine which simulates the vehicle dynamics. The constraints on derivatives itself adds robustness to model error, and the low-level skill policies further adapt to residual uncertainty during training by learning policies to execute the skill safely. When no simple model is available, a differentiable surrogate can be learned and used in the same way. To study more about the influence of modelling errors, we provide **an additional experiment** in the bottleneck environment with introducing up to 20% noise in state observations. The results are summarized in the table below. As can be seen, our proposed approach can learn safe policies in the presence of uncertainties.
>
> | Environment   | Success Rate | Training Steps |
> |---------------|-------------:|---------------:|
> | **Bottleneck**| 0.99 ± 0.03  | 300K |
>
>
> ---
>
> > **Q3**: Five seeds are unusually low for high-variance MARL benchmarks. Please report 10–20 seeds with 95 % confidence intervals, otherwise this severely impacts the significance of the claims.
>
> We ran the experiments with 10 seeds for 300k iterations and present results with 95% confidence intervals. The presented results are consistent with Table 1 in the results section.
>
> | Environment | Success Rate | Avg. Travel Time | Avg. Energy |
> |-------------|---------------:|-------------------:|--------------:|
> | **Merging** | 0.99 ± 0.01 | 155.48 ± 8.32 | 13.01 ± 2.32 |
> | **Bottleneck** | 0.99 ± 0.03 | 232.16 ± 41.19 | 7.46 ± 0.23 |
> | **Roundabout** | 0.99 ± 0.02 | 405 ± 33.34 | 10.00 ± 0.70 |
> | **Tollgate** | 0.99 ± 0.02 | 373 ± 26.36 | 8.79 ± 0.28 |
> | **Intersection** | 0.979 ± 0.03 | 317 ± 47.39 | 5.35 ± 0.53 |
>
> ---
>
> > **Q4**: Please give intuitive/informal explanations in Section 3.2.
>
> Thanks for the suggestion. We will add the following clarification at the beginning of Section 3.2: (HO)CBFs are a class of CBFs that offer sufficient conditions for ensuring safety for systems with higher relative degree. For example, for a point mass with acceleration control, the system's relative degree is 2, and only using a position-based CBF cannot ensure safety because the control term does not appear in the CBF constraint. Therefore, we need to also consider velocity in the CBF, making this CBF a HOCBF.
>
> ---
> ##Weaknesses##
> ---
>
> **Quality**
>
> > **1**. The requirement of knowing affine system dynamics (Eq. 3). and the robustness to modelling error.
>
> Please refer to our reply to **Q2**. In addition, for the study of the robustness to modelling error, the broader robust‑CBF theory provides means for formal treatment of bounded model errors. This is out of the scope of this paper and we leave it to future work.
>
> ---
>
> > **2**. Hand-crafted pre-defined skills.
> Please refer to our reply to **Q1**. Also, our method requires high level task prior to choose an initial skill set, and imposes no restriction on the cardinality or diversity of skills set. Thus, any set of skills can be chosen based on high level task related knowledge.
>
> ---
>
> > **3**. Key baselines such as MAPPO augmented with identical skills and CBF filters are missing.
>
> Please refer to our reply to **Q1**.
>
> ---
>
> > **4**. Comparisons are averaged over only five seeds, and no statistical significance tests are reported.
>
> Please refer to our reply to **Q3**.
>
> ---
>
> **Clarity**
>
> >**1**. Symbols, notations, equation numbers, acronyms, typos and references revision.
>
> Thank you for pointing these out. We will address these comments in the final submission.
>
> ---
>
> > **2**. Section 3.2 offers little intuition for (HO)CBFs.
>
> Please refer to our reply to **Q4**.
>
> ---
>
> > **3**. Clarification on joint state space and agent state and observation space notations.
>
> The joint state space is defined as $\mathcal{S} = \mathcal{S}^{e} \times \prod_{i\in\mathcal{N}} \mathcal{S}^{i}$ where $\mathcal{S}^{e}$ denotes the environment state space and $\mathcal{S}^{i}$ the state space of agent $i$. Each agent $i$ receives an observation $o^{i} \in \mathcal{O}^{i} \subseteq \mathcal{S}$, which includes its own state $s^{i}$ together with those portions of other agents’ or environmental states that are observable to it. Safety is encoded by a control‑barrier function $b : \mathcal{O}^{i} \to \mathbb{R}$; equivalently, one may write $b(s^{i}, s^{j})$ to emphasise the dependence on agent $i$’s state and the observed state $s^{j}$ of another entity. The safety constraint is expressed in the form $b(o^{i}) \ge 0$.
>
> ---
>
> **Significance**
>
> >**1**. Generalization to higher-dimensional platforms (e.g., MuJoCo robots like the Ant) or to settings with learned dynamics is not demonstrated. Until such breadth is shown (or at least argued theoretically) the broader impact on real-world applications remains uncertain.
>
> We require only an approximate dynamics model for the CBF constraints. Emerging differentiable‑simulation frameworks (e.g., JAX + Diffrax, Brax, MJX) can learn such surrogates online, eliminating the manual dynamics‑derivation burden in future. Our experiments on MetaDrive offers manifold benefits, namely: (i) simulate large number of physical agents, (ii) incorporates hard safety‑critical interactions between agents. By contrast, standard MuJoCo multi‑agent set‑ups—Ant, Humanoid, etc.—consist of a single body whose limbs are treated as pseudo‑agents or whose policies are merely distributed; and offers limited scalability of the number of agents. Our traffic scenarios, on the other hand, allows for validating the scalability of our proposed approach, as demonstrated from results specifically table 6.
>
> ---
>
> **Originality**:
>
> > The paper could better differentiate itself from prior safe-HRL works to underscore what is truly unique.
>
> Based on our survey, there are no prior works on Safe Hierarchical Multi-Agent RL (Safe-HMARL). The prior works are primarily focused on HRL and HMARL as stated in the paper. Our work is the first literature on tackling safety using Hierarchical RL for multi-agent systems. There are prior works on hierarchical control [1], and hierarchical policy refinement using adaptive chance constraints for single-agent systems [2] which is distinctly different from Safe-HMARL.
>
> ---
>
> ## References
> [1] L. Zhang, G. Cheng and W. Wang, "Safe Reinforcement Learning-Based Adaptive Hierarchical Control Approach for Virtual Coupling Trains," (ICNSC), 2024.
>
> [2] Z. Chen et al., "Safe Reinforcement Learning via Hierarchical Adaptive Chance-Constraint Safeguards," 2024 IEEE IROS, UAE, 2024, pp. 12378-12385.

---

> ### Author Response · Authors · 2025-08-04
> **Re-evaluation and further discussion**
>
> We will like to thank you once again for the thoughtful comments and feedback. We hope our response has addressed your questions and concerns. We will greatly appreciate if you could share any further questions and comments you have and accordingly re-evaluate our submission based on the response.
>
> Thanking you,  \
> Authors

---

> > ### Author Response · Authors · 2025-08-07
> >
> > Thank you once again for your insightful feedback and comments.
> >
> > Following our response, we were politely reaching out to inquire about your further feedback and queries and will address them promptly. We will be thankful if you could consider revising your score if we have addressed your concerns in our response.
> >
> > Thanking you, \
> > Authors.

---

### Note · Authors · 2025-08-14

We thank the chairs and reviewers for their time, feedback, and dedication to a rigorous review process.

## Strengths
The strengths underlined by the reviewers for our paper are summarized below.

- We present the **first paper** on Safe Hierarchical Multi-Agent RL.
- Our hierarchical approach is **scalable** to a large number of cooperative agents.
- Our method exhibits strong empirical performance, achieving **near perfect success/safety rate** in challenging safety-critical benchmarks.

## Concerns
Below, we summarize our responses to the primary concerns shared by the reviewers:

- Handcrafted skills: Our skills are chosen intuitively based on the high-level knowledge of the task. The only component that is predefined in a skill is its termination condition. All the other components, such as its intrinsic reward and low-level policy, are automatically derived or learned without additional knowledge. For example, in the driving task, a speed-up skill's termination condition is a predefined increment on the current velocity, the corresponding intrinsic reward is a norm of the target velocity error, and the low-level policy is trained jointly with the high-level policy using our novel HMARL-CBF algorithm. Moreover, we show that these learned skills are transferrable to other similar environments.

- Baseline comparisons: We have added results on additional safe MARL algorithms. Together with the paper’s results, we have compared our approach against existing RL, MARL and Safe MARL methods outlined below.

    - RL: IPPO
    - MARL: MAPPO, MAPPO with various penalties, MFPO, CL, CoPo
    - Safe MARL: MAPPO Lagrangian, MACPO, DGPPO

    Our results demonstrate that the proposed HMARL-CBF method significantly outperforms existing approaches in achieving safety and high performance in safety-critical environments with a large number of agents. In addition, we have performed extensive ablation studies including using a uniform high-level policy, random constraint dropout in the low-level policy, hierarchical MARL with LQR as the low-level skill, and handcrafted CBF-based low-level policy.

- Assume exact transition dynamics and does not handle modeling errors: We have clarified that we only use approximate dynamics model (e.g., a bicycle model in the PyBullet-based MetaDrive environment) for implementing the CBF constraints. We further note that the CBF constraints are parameterized and this parameter learning provide robustness against modeling errors.

---

### Decision · Program_Chairs · 2025-09-17

**Decision:**

Accept (poster)

**Comment:**

This paper presents a method for safe hierarchical multi-agent reinforcement learning (MARL) using control barrier functions. The approach learns a higher-level policy over skills for coordination while the lower-level policy executes those skills in a safe manner with control barrier functions.

The approach is novel and performs well in terms of safety and scalability in the experiments.

There were concerns about the experiments, the presentation, and some assumptions (e.g., knowledge of skills and approximate dynamics) but these were mostly addressed during the author response and discussion. The paper should be updated to include this new material.